# The caspase-activated DNase promotes cellular senescence

Aladin Haimovici [ID] [1✉], Valentin Rupp[1], Tarek Amer[1], Abdul Moeed [ID] [1], Arnim Weber [ID] [1] & Georg Häcker [ID] [1,2✉]

## Abstract

**Cellular senescence is a response to many stressful insults. DNA damage is a consistent feature of senescent cells, but in many cases its source remains unknown. Here, we identify the cellular endonuclease caspase-activated DNase (CAD) as a critical factor in the initiation of senescence. During apoptosis, CAD is activated by caspases and cleaves the genomic DNA of the dying cell. The CAD DNase is also activated by sub-lethal signals in the apoptotic pathway, causing DNA damage in the absence of cell death. We show that sub-lethal signals in the mitochondrial apoptotic pathway induce CAD-dependent senescence. Inducers of cellular senescence, such as oncogenic RAS, type-I interferon, and doxorubicin treatment, also depend on CAD presence for senescence induction. By directly activating CAD experimentally, we demonstrate that its activity is sufficient to induce senescence in human cells. We further investigate the contribution of CAD to senescence in vivo and find substantially reduced signs of senescence in organs of ageing CAD-deficient mice. Our results show that CAD-induced DNA damage in response to various stimuli is an essential contributor to cellular senescence.**

**Keywords** Senescence; Caspase-activated DNase; Apoptosis; Ageing
**Subject Category** Autophagy & Cell Death

## Introduction

Cellular senescence is a state characterized by permanent cell cycle arrest, macromolecular damage, metabolic changes and secretion of various cytokines/chemokines, growth factors and metabolites, collectively known as the senescence-associated-secretory phenotype (SASP) (Gorgoulis et al, 2019). Senescence is a response to many stressors, both physiological and toxic, for instance chemotherapy (Schmitt et al, 2022). Senescent cells accumulate during organismal ageing (Childs et al, 2014) and can contribute to numerous age-related pathologies (Bussian et al, 2018; Ferreira-Gonzalez et al, 2018; Mylonas et al, 2021). DNA damage is among the common features of senescence. Senescent cells often exhibit persistent foci of a nuclear DNA-damage response, most typically localized to the telomeres (Gorgoulis et al, 2019). DNA-damaging agents, such as cytotoxic drugs or reactive-oxygen species can induce senescence, and the response to DNA damage is considered a driver of the SASP (Gorgoulis et al, 2019). However, senescence-inducing stresses do not necessarily directly damage DNA, and the source of the identified DNA damage is often unclear.

The caspase-activated DNase (CAD) is a cellular enzyme initially discovered for its activity during apoptosis, when it is activated by cell death caspases and degrades the genomic DNA (Enari et al, 1998; Liu et al, 1998). Later work has, however, established that CAD activity is dispensable for apoptotic cell death: cells lacking CAD activity die normally, and a CAD-deficient mouse has no apoptosis-related phenotype (Nagata et al, 2003). Recent work suggests other roles of CAD in cellular differentiation and in cancer (Benada et al, 2023). Importantly, CAD activation is not necessarily linked to apoptotic cell death. The apoptotic signaling pathway can be activated to a sub-lethal level where some signaling events such as low-level caspase activation are detectable but the cell does not die (Ichim et al, 2015). However, CAD activation appears to be a regular feature of such sub-lethal signaling (Brokatzky et al, 2019; Dorflinger et al, 2022; Ichim et al, 2015). Signs of CAD activity can even be detected in cancer cells growing in the absence of an apoptotic stimulus (Haimovici et al, 2022; Liu et al, 2017); in tumor cell lines in vitro, spontaneous CAD activity is the result of spontaneous, non-lethal activation of the mitochondrial apoptosis apparatus (Haimovici et al, 2022). Spontaneous CAD activity can also be detected in non-transformed mouse intestinal organoids (Haimovici et al, 2022), indicating that it is not limited to malignant cells.

We hypothesized that CAD may be a contributor to senescence. Because non-apoptotic CAD activity appears to be a common event in physiology and pathology, CAD may be a factor responsible for senescence-associated genomic DNA damage. Further, it has been observed numerous times that stimuli that induce apoptosis may also lead to senescence, and that certain, often unidentified cellular characteristics decide about this dichotomous outcome (Childs et al, 2014). One way of interpreting this is that apoptosis and senescence are both triggered by the apoptosis pathway, and that the stimulus strength makes the decision whether the cell dies by

[1]Institute of Medical Microbiology and Hygiene, Medical Center, University of Freiburg, Faculty of Medicine, Freiburg, Germany. [2]BIOSS Centre for Biological Signalling Studies, University of Freiburg, Freiburg, Germany. ✉E-mail: aladin.haimovici@uniklinik-freiburg.de; georg.haecker@uniklinik-freiburg.de

apoptosis or, alternatively, generates only sub-lethal signals that activate CAD which may lead to senescence.

We undertook this study to test for the potential role of CAD in senescence. We observed that CAD was necessary for senescence in response to established triggers, and CAD activation was sufficient for senescence in human cell lines. Ageing CAD-deficient mice had reduced markers of cellular senescence in several organs. We propose that CAD is a broadly acting contributor to DNA damage in the induction of senescence.

# Results

## Sub-lethal activation of CAD can trigger senescence

One prediction of our hypothesis is that mitochondria, through generating sub-lethal signals, can initiate senescence that depends on CAD. We used the BCL-2/BCL-X$_L$-small-molecule inhibitor ABT-737 to test this. ABT-737 neutralizes anti-apoptotic BCL-2 and BCL-X$_L$, triggering the release of cytochrome $c$ and caspase activation especially in cancer cells (Oltersdorf et al, 2005). In susceptible cells, for instance some forms of B-cell malignancy, ABT-737 induces apoptosis. In many cancer cell lines in vitro, ABT-737 at moderate doses causes little overt apoptosis on its own, but it can be used to trigger low-level (sub-lethal) caspase- and CAD activation (Ichim et al, 2015). Indeed, ABT-737 has been found to induce senescence in cancer lines (Song et al, 2011) and human fibroblasts (Victorelli et al, 2023). We treated BJ or WI-38 human fibroblasts, mouse embryonic fibroblasts (MEFs) or MDA-MB-231 human breast cancer cells with ABT-737. Little cell death was induced at the concentrations of the inhibitors used (Fig. EV1A). Fibroblasts were treated for 21 days, replacing ABT-737 every 48 h as reported (Victorelli et al, 2023), while for MDA-MB-231 cells single treatment for 24 h (followed by analysis after 1 week) was sufficient. ABT-737-treated cells showed a typical senescent morphology, and a significant number of ABT-737-treated cells expressed senescence-associated β-galactosidase (SA-β-Gal). Further markers of senescence (loss of proliferation, a DNA-damage response, loss of lamin B1-expression and upregulation of a number of senescence-associated genes) were also observed with slight differences between the cells (Fig. 1A–H). However, the appearance of senescence markers was absent or substantially reduced in cells where CAD had been genomically deleted (Fig. 1A–H, see Fig. EV1B for confirmation of deletion). As expected, ABT-737 treatment caused a DNA-damage response that depended on CAD and that was blocked by caspase inhibition. CAD can, in addition to the proteolytic activation by caspases, be activated by the kinase ATM (Larsen et al, 2022). ATM-inhibition did not alter the appearance of the CAD-dependent DNA-damage response signal (Fig. EV1C). Similar results were observed with two other cell lines (Fig. EV2A–G). These data indicate that sub-lethal signals through the mitochondrial apoptosis pathway can initiate CAD-dependent senescence.

## Triggers of cellular senescence in fibroblasts depend on CAD

Activated oncogenes have senescence-inducing activity. This process, known as oncogene-induced senescence (OIS), restricts proliferation of incipient tumor cells and can therefore be tumor-suppressive. We next tested for a role of CAD during OIS using a constitutively active version of the RAS-oncogene (Schmitt et al, 2022). We transduced wt and CAD-deficient mouse embryonic fibroblasts (MEFs) with a retrovirus driving expression of H-RAS$^{G12V}$. Ten days after transduction, we observed the expected OIS-response, visible as induction of SA-β-Gal activity in wt cells and loss in proliferation, accompanied by increased expression of *p16* and *p21*. The histone methylation mark H3K9me3, another marker of senescence cells, accumulated upon expression of H-RAS$^{G12V}$, the levels of lamin B1 increased and the SASP component IL-6 was increased in the supernatants of H-RAS$^{G12V}$-transduced cells. All markers tested were significantly less induced in CAD-deficient cells, suggesting that RAS-induced OIS is CAD-dependent (Fig. 2A–D). We also tested longer time points after RAS-transduction (for this we had to use human fibroblasts (BJ) because in MEFs replicative senescence kicks in and would interfere). CAD-deficient BJ fibroblasts, 14 days after transduction, also started to show a reduction in proliferation and by 21 days, they had acquired senescent morphology and showed similar growth reduction as wt cells, suggesting that the loss of CAD can be compensated later on at least in these cells (Appendix Fig. S1).

Continuous or repeated signals through type-I interferon (IFNβ) are also well-established drivers of cellular senescence. IFNβ can induce senescence and can also be part of the SASP, propagating senescence in tissues (Frisch and MacFawn, 2020). Daily exposure of MEFs to IFN β over the course of 10 days induced senescence in wt cells as expected and measured by SA-β-Gal activity, reduction in proliferation, the induction of senescence-associated genes and IL-6-secretion. Again, all these features of senescence were significantly reduced in CAD-deficient compared to wt MEFs (Fig. 2E–H). To test whether the role of CAD was the effect of its DNA-damaging activity, we treated cells with the direct DNA-damaging agents $H_2O_2$ (Canli et al, 2017; Imlay et al, 1988) or doxorubicin. There was a small effect of CAD-loss on the induction of IL-6 by $H_2O_2$ but no difference for the other senescence markers tested (Appendix Fig. S2). Doxorubicin-induced senescence in MDA-MB-231 cells as expected. Surprisingly, CAD-deficient cells showed reduced senescence by all parameters tested (Fig. EV3A–C). We repeated these experiments with WI-38 fibroblasts. In these cells, CAD-loss had a minor effect on the number of SA-β-Gal expressing cells but no effect on the other parameters tested (Fig. EV3C–F). Direct DNA damage therefore still can induce senescence in the absence of CAD, suggesting that it is the canonical DNA-damaging function of CAD that is required for senescence. A contribution of CAD was still detectable in some situations, especially in the MDA-MB-231 cells. As mentioned above, CAD can also be activated by the ATM-driven DNA damage response. It seems a likely possibility that ATM-activated CAD can contribute to senescence, and the size of this effect will depend on the precise circumstances. We also added caspase-inhibitor or ATM inhibitor to MDA-MB-231 cells during doxorubicin treatment. There was no effect of the inhibitors on the induction of P21 (Fig. EV3G). While it is difficult to be sure of the effects of inhibitors during such long treatments, it may indicate that disrupting one pathway is not sufficient to block the induction of senescence. The results show that CAD activation is a mediator of senescence in established situations of OIS, of interferon-induced senescence and may contribute to senescence induced by a chemotherapeutic drug.

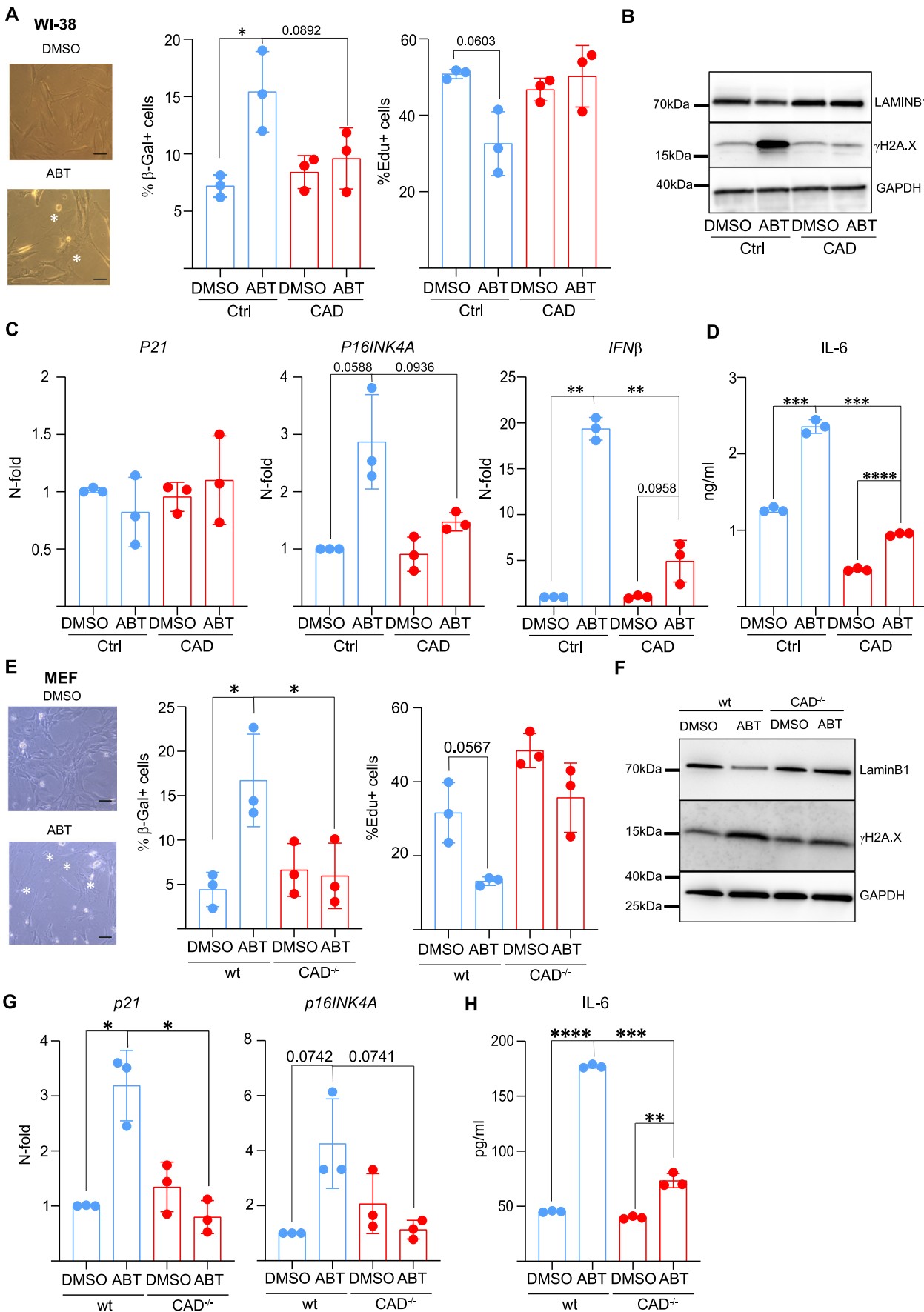

**Figure 1.  Sub-lethal activation of CAD can trigger senescence.**

(A) WI-38 fibroblasts (carrying a non-targeting gRNA (Ctrl) or CAD-deficient) were treated with 5 µM ABT-737. Media was replaced with fresh ABT every 48 h over a period of 21 days. Bright-field images show cells after 21 days of ABT treatment. White stars highlight cells with enlarged and flat morphology. Scale bar: 50 µm. Cells were stained with β-galactosidase staining solution, and Edu and percentages of SA-β-Gal⁺ and Edu⁺ cells were quantified by microscopy. (B) Cells were treated as in (A). Lamin B1 and γH2A.X protein expression was analyzed by western blot. GAPDH was used as a loading control. (C) Expression of senescence-associated genes was measured by RT-PCR. (D) Supernatants from cells cultured for 21 days were analyzed by ELISA for IL-6. (E) MEF (from wild-type or CAD-deficient embryos) were treated as in (A), except the duration of the experiment was 7 days. Bright-field images show cells after 7 days of ABT treatment. White stars highlight cells with enlarge and flat morphology. Scale bar: 50 µm. Cells were stained with a β-galactosidase staining solution, and Edu, and percentages of SA-β-Gal+ and Edu+ cells were quantified by microscopy. (F) Lamin B1 and γH2A.X protein expression were analyzed by western blot. GAPDH was used as loading control. (G) Expression of senescence-associated genes was measured by RT-PCR. (H) Supernatants from cells cultured for 7 days were analyzed by ELISA for IL-6. Each symbol shows the result from one independent experiment. Data represent the mean/SD. An unpaired parametric *t* test (with Welch's correction) was used to calculate statistical significance. *$P < 0.05$, **$P < 0.01$, ***$P < 0.001$, ****$P < 0.0001$. Source data are available online for this figure.

## CAD is an inhibitor of spontaneous immortalization of mouse embryonic fibroblasts

When primary MEFs are replated and cultured for numerous cycles, they reach a growth plateau before some cells can spontaneously "immortalize" and continue proliferation, a phenomenon referred to as replicative senescence (Gorgoulis et al, 2019). Because CAD is active spontaneously in epithelial cells in culture (Haimovici et al, 2022), we considered the possibility that CAD activity also occurs during culture of MEFs and may also play a role in the replicative senescence and in inhibiting spontaneous immortalization of MEFs. To test this, we used the standard 3T3 method of spontaneous immortalization of primary MEFs (Xu, 2005). Primary MEFs were isolated from wt or from CAD-deficient mice, taken into cell culture and replated every 3 days; proliferation was measured by cell counting. The initial proliferation of the cells was similar between genotypes. However, after 30 days of serial passages, wt cells reduced their proliferation rates while CAD-deficient cells continued to proliferate and to increase in number (Fig. 3A). Expression of SA-β-Gal was detectable in wt MEFs as expected but much less was expressed in CAD-deficient MEFs (Fig. 3A). We analyzed the expression of additional senescence markers and found that *p21, p16, Tnf, Mmp3,* and *Mmp13* genes were upregulated in wild-type MEF as expected but not appreciably induced in CAD-deficient MEFs (Fig. 3B). After one month of culture, supernatants from wt MEFs showed substantially increased levels of the SASP factors IL-6 and CXCL10, and this was not seen in supernatants from CAD-deficient MEFs (CCL2, which may also be a SASP factor in some cells, was induced in MEFs from both genotypes) (Fig. 3C). These results suggest that CAD is activated during prolonged culture of MEFs, induces senescence and is a factor limiting spontaneous immortalization in these cells.

## CAD activation is sufficient to induce senescence

DNA damage can be a stimulus for senescence. Gamma-irradiation for instance has been found to cause a DNA-damage response (DDR) and consecutively senescence. For γ-irradiation a dose threshold has been described: at low energies, the DNA-damage response (DDR) was transient and did not trigger senescence, while high doses of irradiation induced a persistent DDR and senescence. This senescence response was intimately linked to the production of the SASP factor IL-6 (Rodier et al, 2009). CAD is activated by upstream signals and appears to be required for senescence. We tested whether isolated activation of CAD, and the DNA damage it causes, may also be sufficient for the induction of senescence. We

used the human keratinocyte HaCaT cell line where we have established a system for targeted destruction of the CAD-inhibitor ICAD by fusion to a small protein tag, an auxin-induced degron (AID; HaCaT-ICAD-mAID-GFP cells) (Haimovici et al, 2022; Nishimura et al, 2009). Auxin treatment of these cells causes the proteasomal degradation of ICAD and the activation of CAD (Haimovici et al, 2022). Single, one-time activation of CAD was insufficient to cause senescence. Because the level of DNA damage is likely to determine the decision to initiate the senescence program, we established a treatment protocol where we repeatedly stimulated HaCaT-ICAD-mAID-GFP cells with auxin to activate CAD over a 2-week period (Fig. 4A). At that stage, cells started to show morphological changes characteristic of a senescent phenotype: they became flat, enlarged, expressed SA-β-Gal activity and showed reduced proliferation (Fig. 4B) and a reduction in lamin B1 levels (Fig. 4C). Gene-expression analysis identified the upregulation of *P21* and the SASP genes *IFNβ* and CXCL10 (Fig. 4D). The cells secreted higher levels of IL-6 (Fig. 4E). We also introduced an auxin-inducible CAD into WI-38 fibroblasts. Early experiments suggest that CAD activation also induces senescence in these cells (Appendix Fig. S3A).

Micronuclei and cytosolic chromatin fragments (CCF) are both cytosolic DNA-containing structures that are released from the nucleus in certain circumstances; CCF are a regular feature of senescence (Gorgoulis et al, 2019). CAD activity can generate micronuclei (Haimovici et al, 2022; Ichim et al, 2015). We first tested whether CAD can also generate CCF and found that upon CAD activation, cytosolic DNA-structures are detectable that are either positive for the CCF-markers H3K27me2me3 or H3K9me3 (Miller et al, 2021) or negative for these markers (presumably micronuclei; Appendix Fig. S3B). Cytosolic DNA can be recognized by the cytosolic DNA-receptor cGAS, which activates STING and can cause the secretion of cytokines, including SASP factors (Harding et al, 2017; Mackenzie et al, 2017). cGAS has recently been reported to be required for the induction of the SASP upon the recognition of mitochondrial DNA (Victorelli et al, 2023). We tested whether cGAS also contributed to the induction of senescence downstream of CAD activation.

We deleted cGAS in the auxin-inducible HaCaT cells (Moeed et al, 2024) and subjected them to the same senescence-inducing protocol. CAD activation induced the expression of SA-β-Gal, reduced proliferation and the expression of lamin B1 as well as the induction of p21 in a way comparable to the control cells (Fig. 4B–D). However, the induction of SASP factors (mRNA for IFN-β and CXCL10 and secretion of IL-6) was absent or strongly reduced in cells lacking cGAS (Fig. 4D,E). The results indicate that

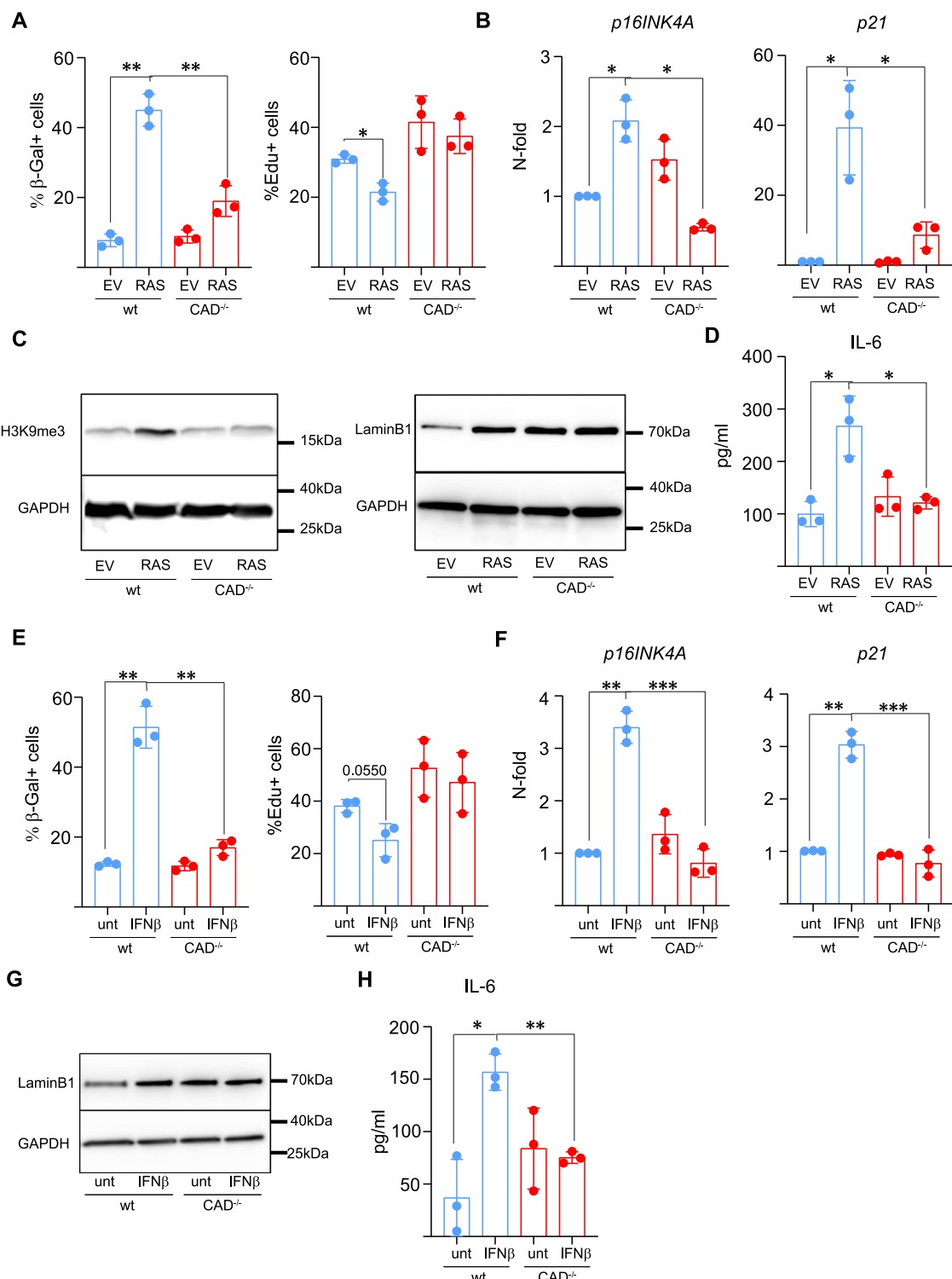

cGAS has a function in inducing the SASP genes downstream of CAD but is otherwise dispensable for the CAD-dependent induction of senescence.

The SASP has effects on the immune system, for instance by attracting myeloid cells, and in tissue remodeling and wound healing. It has further established functions in reinforcing and spreading senescence in a paracrine and autocrine fashion (Gorgoulis et al, 2019).

We therefore tested whether the soluble factors produced by senescent HaCaT cells upon isolated, repeated CAD activation, also had such paracrine effects. We incubated BJ human fibroblasts with conditioned medium of HaCaT cells expressing active CAD. After 1 week of culture with the conditioned medium, BJ cells showed reduced proliferative activity, signs of a DDR and loss of lamin B1 (Figs. 4F–H and EV4A). These results suggest that isolated activation of CAD is sufficient to

**Figure 2.  Induction of senescence by oncogenic RAS, and type-I interferon depends on CAD.**

(A–D) wt and CAD-deficient MEFs were transduced with empty retrovirus or pLNCX2-ER:H-RAS$^{G12V}$-virus and stimulated each day with tamoxifen to activate H-RAS$^{G12V}$. (A) Cells were stained with β-galactosidase and Edu staining solution and percentages of SA-β-Gal$^+$ and Edu$^+$ cells were quantified by microscopy after 10 days in culture. (B) Expression of senescence-associated genes 10 days after retroviral transduction. Gene expression was analyzed by RT-PCR and normalized to GAPDH. Relative expression compared to wt cells transduced with empty vector is shown. (C) Cells were lysed and western blot for H3K9me3 and Lamin B1 was performed. GAPDH was used as loading control. (D) Supernatants from cells cultured for 10 days post-transduction were analyzed by ELISA for IL-6. (E–H) wt and CAD-deficient MEFs were treated daily with 10 ng/ml of IFN-β for 10 days. (E) Cells were stained with β-galactosidase and Edu staining solution and percentages of SA-β-Gal$^+$ and Edu$^+$ cells were quantified by microscopy after 10 days in culture. (F) Expression of senescence-associated genes were analyzed after 10 days in culture by RT-PCR. The expression relative to untreated wt cells is shown. (G) Cells were lysed and western blot for Lamin B1 was performed. GAPDH was used as loading control. (H) Supernatants from cells cultured for 10 days were analyzed by ELISA for IL-6. Data are the mean/SD of 3 independent experiments. Each symbol represents one experiment. Unpaired parametric *t* test (with Welch's correction) was used to calculate statistical significance. *$P < 0.05$, **$P < 0.01$, ***$P < 0.001$. Source data are available online for this figure.

induce a senescent phenotype, and soluble factors produced as part of the DDR appear to contribute in an autocrine and/or paracrine fashion.

## A CAD-induced DDR at telomeres

Telomeres play a prominent role in senescence. Telomere shortening occurs during cell division and is believed to be associated with replicative senescence. A DDR at telomeres is further associated with many forms of senescence (Hewitt et al, 2012; Victorelli and Passos, 2017). There is no evidence that CAD cleaves genomic DNA specifically at telomeres. CAD is a non-specific nuclease but it has a wide scissor-like structure that does not permit it to cleave DNA that is bound by proteins (Woo et al, 2004). During apoptosis, CAD-dependent cleavage of genomic DNA primarily between nucleosomes has been observed (Enari et al, 1998; Wyllie, 1980); cleavage in the internucleosomal linker DNA has been confirmed with recombinant protein and isolated nuclei (Widlak et al, 2000). A recent study investigating CAD activity upon its activation by DNA damage has found the introduction of single-strand breaks adjacent to linker histones but throughout the genome (Larsen et al, 2022). The main constraint on the activity of CAD therefore appears to be chromatin structure.

Despite this lack of specificity in CAD-dependent DNA cleavage, there is evidence that the CAD-dependent DDR has a preference for telomeres: during prolonged mitotic arrest (a situation triggering sublethal signals in the apoptosis pathway (Orth et al, 2012)), a caspase- and CAD-dependent DDR, especially at telomeres has been described (Hain et al, 2016). We therefore tested for telomere-associated DDR foci (TAF), an established marker of telomere dysfunction and senescence, upon auxin-mediated activation of CAD. We activated CAD with auxin for 6 h then co-stained the cells for telomeres using a specific probe and for the DDR-protein 53BP1. As shown in Fig. EV4B,C, CAD activation induced a significantly increased number of TAF, defined as localization of 53BP1 to telomeres. Although far from exclusive, this indicates a preference of the DDR at telomeres in cells expressing active CAD. Intriguingly, already 6 h after the beginning of CAD activation, there was a detectable reduction in telomere length (Fig. EV4D). These results suggest a particular vulnerability of telomeres to CAD-activation, which may be linked to the senescence-inducing activity of CAD.

## CAD contributes to ageing-associated cellular senescence in mice

Signs of cellular senescence can be found in ageing organisms, where senescence is thought to contribute to the ageing process and to ageing-associated diseases (Zhang et al, 2022). "Spontaneous" activity

of the mitochondrial apoptosis pathway and of CAD occurs at least in proliferating cells in vitro (Haimovici et al, 2022). We therefore considered the possibility that CAD is also activated in tissues in the organism during normal life, where its activation may be associated with cellular senescence. We compared several organ tissues of wt and CAD-deficient mouse littermates, both at young (8–10 weeks) and old (75 weeks) ages for signs of senescence. Kidneys of old wt animals showed substantial areas of SA-β-Gal activity; this area was considerably smaller in tissue from old CAD-deficient animals (Fig. 5A). The number of cells positive for nuclear lamin B1 was higher in kidney sections of old CAD-deficient compared to wt mice, consistent with reduced loss of this marker, and less cellular senescence, in CAD-deficient mice. At the same time, more cells in old CAD-deficient than in wt kidneys were positive for the proliferation marker Ki67; proliferative arrest is one feature of ageing-induced senescence suggesting that CAD activity contributed to this process (Fig. 5B). We tested for the expression of senescence-associated genes in kidney tissue and found the expected upregulation in aged wt mice; this upregulation was strongly reduced, sometimes not detectable in CAD-deficient mice of the same age (Fig. 5C). Senescent cells were also found in the intestinal crypts of old wt mice, as defined by expression of SA-β-Gal activity (Fig. 5D). The intestinal crypts of old CAD-deficient mice contained substantially more cells positive for Ki67 than matched tissue from wt animals (Fig. 5E). Intestinal tissue from old wt mice displayed enhanced expression of senescence-associated genes and this was again much reduced by comparison in old CAD-deficient mice (Fig. 5F). We isolated intestinal crypts from old mice and found that the epithelium of CAD-deficient animals formed more and larger organoids compared to wt (Fig. EV5A,B). From the organoids that did form, more wt organoids expressed SA-β-Gal activity compared to CAD-deficient organoids (Fig. EV5C). Reduced expression of senescence markers (SA-β-Gal activity, expression of senescence-associated genes) was also observed in white adipose tissue (Fig. EV5D,E) and in liver sections (Fig. EV5F) from old CAD-deficient compared to old wt mice. Other senescence genes were not or variably regulated upon ageing in these tissues (Appendix Fig. S4). This analysis of aged mice strongly suggests that CAD is active in normal tissue during the life of a mouse, and that it makes a substantial contribution to cellular senescence during ageing. Together with the results shown above, the data indicate an essential contribution of CAD to numerous instances of senescence.

## Discussion

The results of this study show that CAD activity both is required for many instances of senescence and can be sufficient for

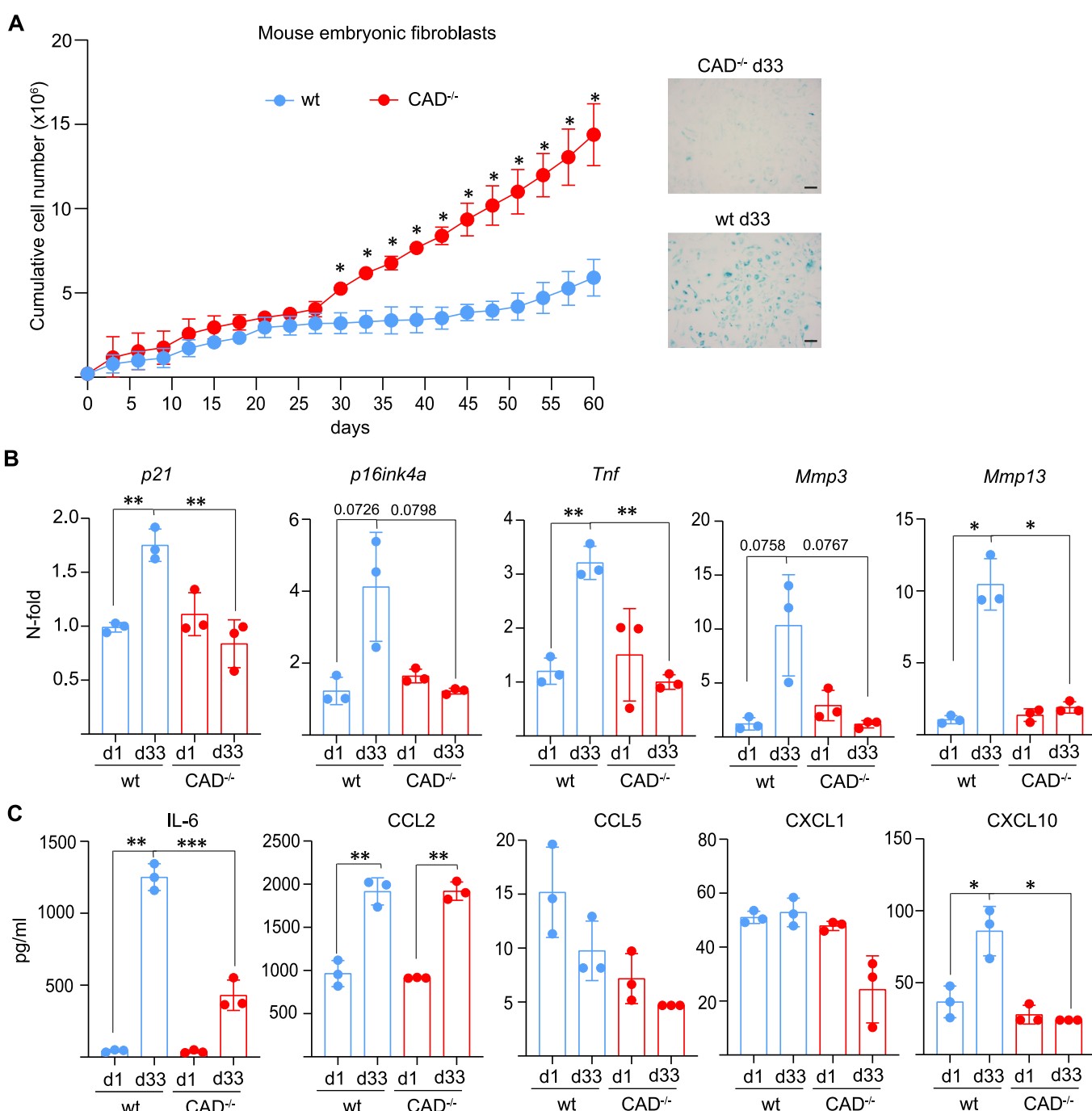

**Figure 3. Deficiency in CAD accelerates spontaneous immortalization of mouse embryonic fibroblasts.**

(A) Cell numbers of cultures of wild-type (wt) and CAD-deficient MEFs. Cells were counted and replated at the same cell number every 3 days, and cumulative cell numbers were calculated. Significant growth differences were detected after about 30 days of culture. Numbers are means/SD of three MEF preparations from individual mice for each genotype. Pictures show cells stained for β-galactosidase after 33 days of culture (scale bar: 50 µm). (B) Expression of senescence-associated genes analyzed at the start of the experiment (day 1) and after 33 days in culture. mRNA was extracted and gene expression was analyzed by RT-PCR. Gene expression was normalized to day 1 wt MEFs. (C) Supernatants from cells cultured from 33 days were analyzed by bead array for IL-6, CCL2, CCL5, CXCL1, and CXCL10. Data are from three pairs of MEFs derived from individual littermate embryos that were tested in parallel. Data are means/SD of three MEF preparations from individual mice for each genotype. Data represent means/SD. Unpaired parametric t test (with Welch's correction) was used to calculate statistical significance. *P < 0.05, **P < 0.01, ***P < 0.001. Source data are available online for this figure.

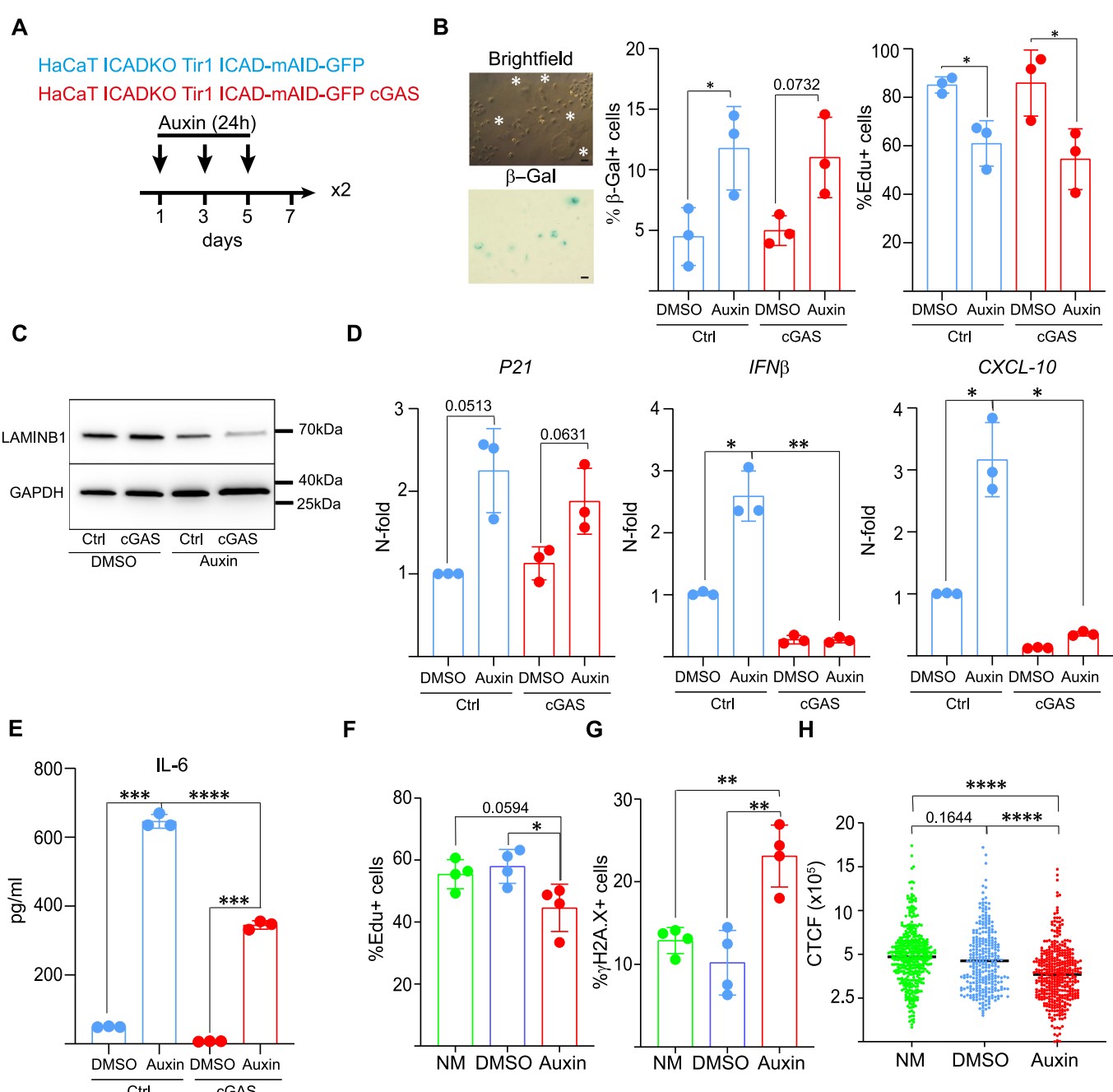

**Figure 4. Direct, isolated CAD activation leads to senescence in HaCaT cells.**

(A) Treatment protocol to induce senescence in HaCaT-ICAD-mAID-GFP and HaCaT-ICAD-mAID-GFP cGAS-deficient cells. Stimulation with auxin (10 μM, 24 h) was conducted for 24 h every second day for 2 weeks, with 2 days recovery after the third stimulation. Cells were harvested 2 days after the last auxin treatment. (B) Representative microscopy pictures of HaCaT-ICAD-mAID-GFP cells after 2 weeks of auxin treatment. Cells were stained with β-galactosidase staining solution. Top panel shows a bright-field picture and bottom panel shows SA-β-Gal+ cells. Scale bar: 10 μm. Flat and enlarged cells (senescent phenotype) are marked by stars. SA-β-Gal+ and Edu+ cells were quantified by microscopy. Each symbol represents one separate experiment. (C) Lamin B1 protein expression was analyzed by western blot. GAPDH was used as loading control. (D) Gene expression was determined by PCR following the auxin treatment protocol in (A). Expression of *P21*, *CXCL10*, and *IFNβ* are shown. Expression is given as fold induction by auxin treatment. Each symbol represents one separate experiment. (E) Supernatants from the cells in (D) were analyzed by ELISA for IL-6. Each symbol represents one separate experiment. (F–H) BJ human fibroblasts were incubated for 7 days with conditioned medium from HaCaT-ICAD-mAID-GFP cells subjected to the protocol in (A) or exposed to solvent (DMSO). Normal medium (NM) was used as a control. (F) Percentage of proliferating (Edu+) cells was quantified by microscopy from at least 500 cells per group. Each symbol represents one separate experiment. (G) γH2AX+ cells were quantified by microscopy from at least 500 cells per group. Each symbol represents one separate experiment. (H) Expression of lamin B1 was determined by immunofluorescence staining and analysis by confocal microscopy (CTCF, corrected total cell fluorescence of lamin B1-stain). Each symbol represents one separate experiment (B, D–G) or one cell (H). Data are the means/SD. An unpaired parametric *t* test (with Welch's correction) was used to calculate statistical significance. *P < 0.05, **P < 0.01, ***P < 0.001, ****P < 0.0001. Source data are available online for this figure.

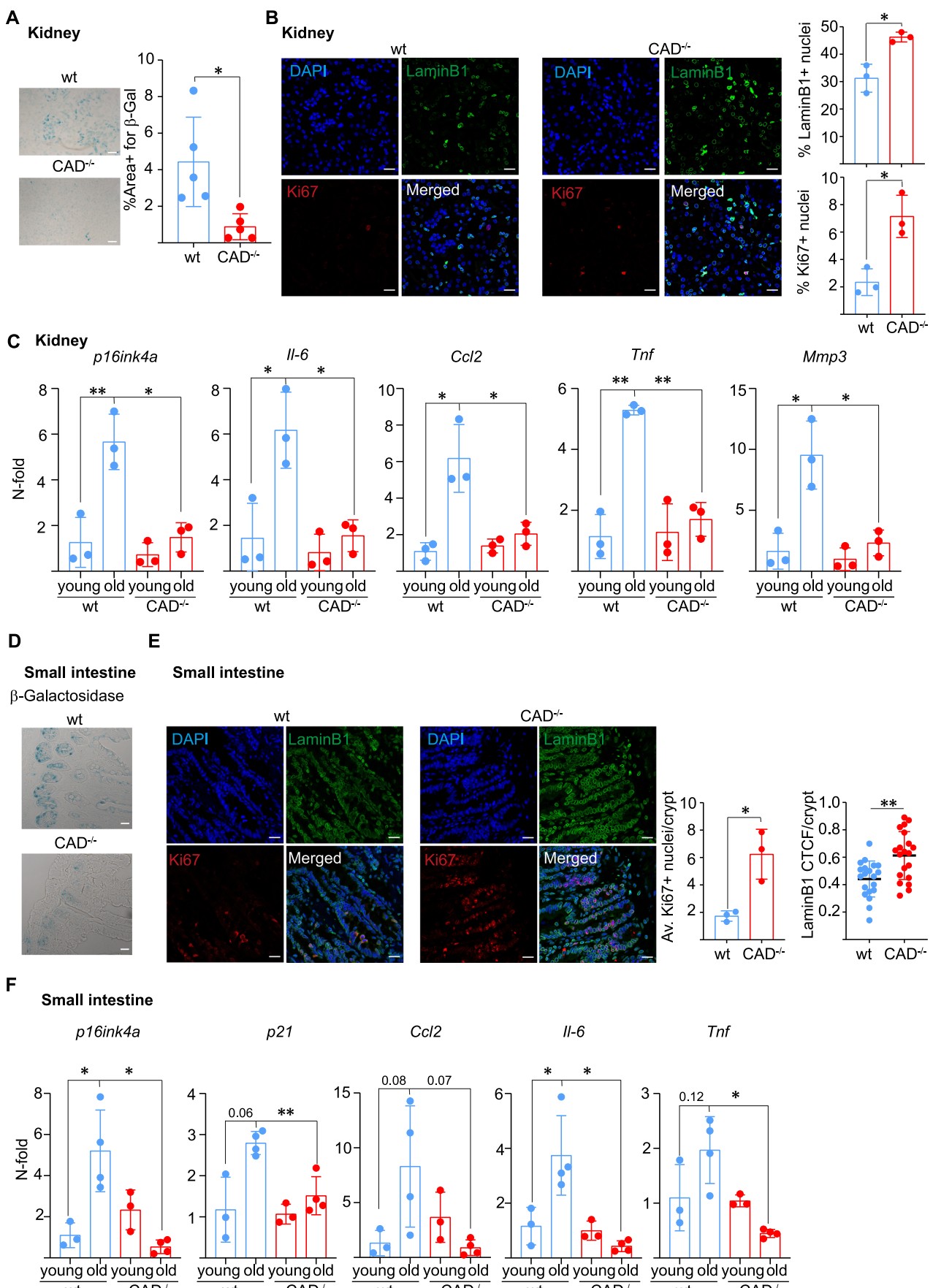

**Figure 5.   CAD-deficient old mice show less markers of senescence in tissues than wt mice.**

(A) SA-β-Gal expression in kidney cryosections from 75-week-old wt and CAD-deficient mice. Left, example of the microscopy (scale bar: 200 μm); right, quantification of the stain in five animals per group. The blue area was quantified by ImageJ. Each symbol represents one animal. (B) Percentage of Lamin B1- and Ki67-positive nuclei in kidneys of 75-week-old wt and CAD-deficient mice. Left, confocal microscopy pictures of kidney cryosection stained for Lamin B1 (green), Ki67 (red) and DAPI (blue). Right, quantification of Lamin B1- and Ki67-positive nuclei. Analysis was performed with ImageJ. Scale bar: 10 μm. (C) Expression of senescence-associated markers in kidney tissues. Gene expression was measured by RT-PCR. Kidney of 8–10-week-old animals (young) (*n* = 3) and 75-week-old animals (old) (*n* = 3) were analyzed. Each symbol represents one mouse. Data were normalized against young wild-type (average of 3 animals) with GAPDH used as a housekeeping gene. (D) Small intestine cryosections from 75-week-old wt or CAD-deficient mice were stained for expression of SA-β-Gal. Microscopic pictures of exemplary sections are shown. Scale bar: 20 μm. (E) Analysis of Lamin B1 degradation and Ki67 in small intestinal crypt of 75-week-old wt and CAD-deficient mice. Left, confocal microscopy pictures of small intestine cryosection stained for Lamin B1 (green), Ki67 (red) and DAPI (blue). Right, quantification of Ki67-positive nuclei per crypt and Lamin B1 CTCF per crypt. Analysis was performed with ImageJ. Scale bar: 10 μm. (F) Expression of senescence-associated genes in small intestine tissue samples. Gene expression was measured by RT-PCR and normalized as in (C). Young (8–10 weeks) (*n* = 3) and old (75 weeks) (*n* = 4) animals were investigated. Each symbol represents one mouse. Data represent the mean/SD. Unpaired parametric *t* test (with Welch's correction) was used to calculate statistical significance. *$P < 0.05$, **$P < 0.01$. Source data are available online for this figure.

senescence induction in human cells. CAD appears to be active in normal mouse tissues, where it drives ageing-associated senescence. These results establish CAD, and CAD-inflicted DNA-breaks, as a key player of cellular senescence and a potential contributor to ageing.

DNA damage is a long-known and well-established factor contributing to senescence, and DNA damage (or a DDR) can typically be observed in the development of senescence (Gorgoulis et al, 2019). There are numerous possibilities how DNA damage can be caused, and DNA damage is exceedingly common (Schumacher et al, 2021). A contribution of CAD to senescence seemed a plausible possibility, because CAD appears to be active in many situations. The reported alternative responses of a cell to the same stimulus, namely to undergo either apoptosis or senescence, can also be seen as suggestive: if a signal threshold is crossed, mitochondrial signals activate enough caspases to kill the cell. Below this threshold, the signals may still activate CAD but lead to senescence. Treatment with the BCL-2-inhibitor ABT-737 illustrates this connection. At the same time, senescent cells appear to be "primed for death", a concept well-established in the cancer field (Certo et al, 2006). This has the consequence that senescent cells are more susceptible to BCL-2-antagonists like ABT-737 (respectively the orally bioavailable version ABT-263 (Chang et al, 2016)) and also to death receptor signaling (Wang et al, 2022). At least in terms of mitochondrial apoptosis, this may well be a consequence of sub-lethal signaling, which would engage the BCL-2-system in a sub-lethal fashion and lower the threshold to apoptosis induction.

Apoptosis can be triggered almost notoriously easily, and it is a reasonable assumption that sub-lethal signals are initiated at least as easily. Our data show that there is a signaling pathway from the mitochondrial apoptosis apparatus, initiated by BCL-2-antagonism, through CAD to senescence. ABT-737 has previously been reported to be able to induce caspase-dependent senescence (Song et al, 2011), and similar caspase-dependent activities have been found for the death receptor CD95 (Raats et al, 2017). It is well-established that mitochondrial signals activate CAD through caspases. It seems likely that many stimuli, like IFNβ and RAS, cause senescence though this pathway. Although they have other signaling qualities, both interferon (Kotredes and Gamero, 2013) and oncogenic RAS (Kauffmann-Zeh et al, 1997) have the potential to induce apoptosis. Sub-lethal signals are probably also generated by these stimuli, and this will activate CAD to drive senescence.

Doxorubicin, which intercalates in and directly damages DNA, and $H_2O_2$, which causes direct oxidative DNA damage, showed no dependency on CAD in fibroblasts. This indicates that it is indeed DNA-damage by CAD that is required; if CAD is in this way bypassed, the chemical agents can induce CAD-independent senescence. Surprisingly, in MDA-MB-231 cells doxorubicin also depended on CAD for senescence induction. There are two plausible explanations for this. First, it is possible that the process involves sub-lethal mitochondrial signals. Chemotherapeutic agents induce apoptosis and very likely at sub-lethal doses sub-lethal signals, which will activate CAD. Secondly, this may work through the recently reported alternative way of CAD activation. Following irradiation, a caspase-independent wave of CAD activation was observed in cancer cells, which depended on the phosphorylation of ICAD by ATM/ATR (Larsen et al, 2022). It seems conceivable that this mechanism also operates when DNA is damaged: the DNA damage caused by secondary CAD activation may be required for senescence. $H_2O_2$-treatment, another stimulus that directly damages DNA, induced senescence independently of CAD. This may however only be a matter of the specific conditions we used.

Isolated CAD activation triggered the secretion of SASP factors, and supernatants from cells expressing active CAD induced signs of senescence, suggesting that CAD can, through the SASP, spread senescence in a paracrine fashion. An important factor of SASP-generation are cytosolic chromosomal fragments/micronuclei (CCF), which can be recognized by cellular recognition systems (especially by the cGAMP synthase, cGAS), leading to the production of IFN and other inflammatory factors (Dou et al, 2017; Gorgoulis et al, 2019). We (Haimovici et al, 2022) and others (Ichim et al, 2015) have reported previously that CAD activation can cause the appearance of cytosolic DNA; the data reported here suggest that both CCF and micronuclei are generated. DNA-recognition by cGAS downstream of CAD activation seems a contributor to the generation of the SASP. Recent work reports that mitochondrial DNA, through cGAS-activation, has a role in inducing the SASP but not other features of senescence (Victorelli et al, 2023). Our results suggest that cGAS is similarly involved in the SASP but not other pathways of senescence upon CAD activation.

At the same time, this model of auxin-dependent CAD activation recapitulates both direct and SASP-dependent DDR and senescence induction. As shown for irradiation-induced DNA damage, the transient DDR caused by low doses of irradiation does not lead to senescence while a persistent DDR, induced by higher levels of DNA damage, does (Rodier et al, 2009). CAD-dependent

senescence was induced by repeated but not by single CAD activation, providing another example of this threshold concept. The CAD-dependent DDR occurred with significant preference at telomeres. Given the preferential cleavage of CAD at internucleosomal junctions (Larsen et al, 2022; Widlak et al, 2000)—where DNA is accessible— this is unexplained although consistent with the previous description of a preferential CAD-dependent DDR at telomeres (Hain et al, 2016). Even when cells are exposed to γ-irradiation, the persistent, senescence-associated DDR preferentially localizes to telomeres (Hewitt et al, 2012). Whether this is the result of less efficient DNA-damage repair at telomeres, or whether the persistence of the DDR at telomeres has other molecular reasons, is unclear.

CAD-deficient mice had substantially fewer senescent cells in several tissues; intestinal stem cells in CAD-deficient mice were better at preserving the potential for organoid formation during ageing. This is intriguing because it indicates that CAD is "spontaneously" active, and that—most likely—sub-lethal signals are constantly generated during normal life. In vitro data suggest that such spontaneous signaling happens at least in some cells (Haimovici et al, 2022). In vivo, there is substantial evidence that cell death caspases can be active (and are active during normal life) in the absence of cell death (McArthur and Kile, 2018). Given that the caspase-dependent activation of CAD appears to occur in all instances of sub-lethal signaling, it is a likely proposition that these instances of sub-lethal caspase activation will also activate CAD. Eventually, this signaling may lead to CAD-dependent senescence.

Although the association may be complex, there is a relationship of DNA damage with both ageing and senescence (reviewed in (Schumacher et al, 2021)): removal of treatment-induced senescent cells on a progeroid background could reduce ageing in mice (Baker et al, 2011), and removing senescent cells can have beneficial effects in atherosclerosis (Childs et al, 2016) and osteoarthritis (Jeon et al, 2017). During normal ageing, chronic DNA damage occurs in numerous organs and tissues (Schumacher et al, 2021). It seems plausible that CAD makes a contribution to these processes. CAD-deficient mice have a normal lifespan in our SPF-facility. Preliminary data further suggest that some of the behavioral changes in ageing mice may be less pronounced in CAD-deficient animals. This point will need more thorough investigation.

Our data identify CAD as a factor contributing to senescence. Both CAD itself and the CAD-activating pathways can in principle be targeted. It may be worth considering such approaches for treating diseases where senescence plays a role.

# Methods

## Cell lines and culture conditions

Mouse embryonic fibroblasts (MEFs) were generated from wild-type and CAD-deficient embryos and cultured in DMEM (Thermo Fisher Scientific, 41965062) with 10% FCS (Anprotec) and 50 μM 2-mercaptoethanol. HaCaT keratinocytes, MDA-MB-231 breast cancer cells (ATCC Cat# HTB-26) were cultured in RPMI (Gibco) supplemented with 10% FCS (Anprotec) and 1% (v/v) penicillin/streptomycin (Gibco). BJ (from ATCC Cat# CRL-2522) and WI-38 fibroblasts (gift from Prof. Clemens Schmitt (Berlin) were cultured in DMEM supplemented with 10% FCS (Anprotec), 1% sodium

pyruvate, 1% glutamax and 1% (v/v) penicillin/streptomycin (Gibco). Auxin (3-indoleacetic acid, Sigma, I2886) was dissolved in DMSO as a 50 mM stock solution. The generation of HaCaT keratinocyte with an auxin-degradable ICAD-mAID-GFP has been described (Haimovici et al, 2022). To activate CAD in WI-38, cells with a 5-Ph-IAA-degradable ICAD-mAID-GFP construct were generated as described for the HaCaT cells with some changes: as with the Hacat cells, ICAD was first knocked-out by CRISPR/Cas9 and selected for puromycin resistance. In a second step, Tir1 was mutated to Tir1-F74G to recognize 5-Ph-IAA (Yesbolatova et al, 2020). Tir1-F74G was then fused to murine ICAD-mAID-GFP via a T2A site and cloned into pENTR1A no ccDB (w48-1) (gift from Eric Campeau and Paul Kaufman; Addgene plasmid # 17398) to generate pENTR1A-TIR1-F74G-T2A-mICAD-mAID-GFP. The construct was then cloned into a lentiviral plasmid (pEF1α-GW-Tir1-F74G-T2A-mICAD-mAID-GFP-puro). Cells expressing GFP were sorted using flow cytometry. Cells were treated with 1 μM 5-Ph-IAA (5-phenyl-1H-indole-3-acetic acid, Cayman Chemical #38161). All cells were cultured at 37 °C in 5% CO2. Gene-deficient cells were generated by CRISPR/Cas9 genome editing by transducing the cells with the lentiviral vector lentiCRISPR v2 (Addgene (Sanjana et al, 2014)) and selection with puromycin (Invitrogen). Guide RNAs were a non-targeting control (ATCGTTT CCGCTTAACGGCG), targeting CAD (TCGGCGTTGTCGGGAA-CACT) or cGAS (ATGATATCTCCACGGCGGCG) as reported (Brokatzky et al, 2019). Cell lines were tested negative for mycoplasma contamination using two different kits: Venor®GeM Mycoplasma Detection Kit for conventional PCR version 1.4 and Lonza MycoAlert™ mycoplasma detection kit.

## SA-β-galactosidase (SA-β-gal) staining

Staining was performed using the kit from Cell Signaling Technologies (#9860) according to the manufacturer's instructions. Briefly, samples were fixed using reagents provided before staining at 37 °C. Whole mount staining on white adipose tissue was performed as described (Baker et al, 2011) with the modification of an 8 h incubation time. For the kidney, small intestine and liver, frozen sections were air-dried before 16 h incubation in staining solution. MEF, MDA-MB-231, BJ, WI-38, and HaCaT cells were counterstained with DAPI, and results are expressed as percentages of SA-β-gal⁺ cells. For kidney and liver sections, images were taken at ×10 magnification. For small intestine sections, images were taken at a ×20 magnification. Percentage positivity (blue area) was calculated over multiple random fields of view using ImageJ. For detection of SA-β-gal activity using flow cytometry, cells were incubated for 1 h with 5 nM Concanamycin A and then further incubated 30 min with SPiDER-βGal (Dojindo, SG02) before analyzing with a flow cytometer or a fluorescence microscope.

## Mice

Mice were on a C57BL/6 genetic background. Ages ranged from 8 to 10 weeks for young to 75 weeks for old mice. Mice were bred and maintained at the animal house of the Freiburg University Medical Center in SPF-conditions. CAD-deficient mice were kindly provided by Prof. Roberto Caricchio, Philadelphia with the permission of Prof. Shigekazu Nagata, Osaka. All mice procedures

were performed in accordance with the relevant ethical guidelines and regulations (authorization number X-23/03K).

## Organoid isolation and culture

Small intestinal crypts were isolated from wild-type and CAD-deficient mice as per protocol by StemCell Technologies. After harvesting, intestinal crypts were kept in Cultrex Matrigel (Bio-Techne, BME001). The culture medium was based on DMEM/F12 + HEPES (StemCell Technologies, 36254) supplemented with B27, N2 supplements (Life Technologies, 17504044, 17502048) and 1.25 μM N-acetylcysteine (Bio-Techne, 5619). R-spondin 1 (conditioned medium), 100 ng/ml murine recombinant Noggin (StemCell Technologies, 78061) and 50 ng/ml murine recombinant EGF (StemCell Technologies, 78016.1) were separately added. Organoids were split every 5 days.

## Histology, immunofluorescence staining, and western blotting

Organs were harvested and fixed in 4% paraformaldehyde for 2 h on ice. Samples were then transferred to a 30% sucrose solution and incubated overnight. Organs were embedded in OCT Tissue Tek (Sakura) and frozen in liquid nitrogen. Samples were stored at −80 °C. For analysis, 5-μm sections were cut using a CM1850 Cryotome (Leica). HaCaT, BJ or MEF cells were incubated on IBIDI slides, fixed with 4% PFA for 10 min, followed by a 10 min permeabilization with PBS containing 0.5% Triton-X. Samples were blocked with PBS/3% BSA and incubated overnight at 4 °C with rabbit anti-53BP1 (1:200) (Abcam, ab36823), rabbit anti-γH2A.X (1:400) (Cell Signaling #9718), rabbit anti-H3K9me3 (1:200) (Abcam, ab8898), mouse anti-H3K27me2me3 (1:200) (ActiveMotif, 39536) or rabbit anti-Lamin B1 (1:1000, Abcam, ab16048) antibodies in PBS/1% BSA. Secondary antibody-solution anti-rabbit-Alexa-488,1:500 (Dianova, 711-545-152) or anti-mouse-Cy5, 1:500 (Dianova, 715-175-151) in PBS/1%BSA, was added for 1 h at room temperature. If needed, samples were incubated 2 h at room temperature with anti-Ki67-Alexa647 (1:100) (Cell Signaling #12075). Samples were stained with DAPI for 5 min and mounted in IBIDI mounting media. For IF on cryosections, slides were permeabilized and blocked 1 h with PBS/1% BSA/0.5% Triton-X-100. After incubation with antibodies, slides were dried at room temperature and mounted on IBID mounting media. Slides were images using a confocal microscope (Zeiss, LSM-710). For western blotting, cells were lysed with RIPA buffer in the wells. Samples were sonicated and heated to 95 °C before loading to SDS PAGE. PVDF membranes were blocked with 5% milk. Proteins were detected with ECL substrate.

## Telomere ImmunoFISH

After 53BP1 staining, slides were washed with PBS, cross-linked with 4% paraformaldehyde for 5 min, incubated with RNAse (100 μg/ml) solution, and dehydrated in graded ethanol. Samples were denatured for 10 min at 85 °C in hybridization buffer (60% formamide (Sigma), 20 mM $Na_2HPO_4$ (pH 7.4), 20 mM Tris (pH 7.4), 2 × SSC and 10 μg/ml salmon sperm DNA (Thermo Fisher, 15632011) containing a Cy-3-labeled telomere-specific (CCCTAA) peptide nucleic acid probe (Panagene, F1002), followed by

hybridization for 2 h at room temperature in the dark. Slides were washed twice in 2×SSC with 0.1% Tween-20 at 57 °C for 10 min, followed by 5-min incubation in 2×SSC containing DAPI. Samples were mounted and imaged using a confocal microscope (Zeiss, LSM-710). Relative telomere length was measured by telomere intensity per nucleus in one z plane as described (Jurk et al, 2014).

## Corrected total cell fluorescence (CTCF)

Following anti-lamin B1-staining, a minimum of 300 cells per condition were imaged using a confocal microscope. Images were taken using a ×63 lens with Z-stacks. CTCF was then calculated with ImageJ using the following formula: CTCF = integrated density − (area of cell × mean fluorescence of background readings). For Lamin B1 quantification on intestinal cryosection, Lamin B1 CTCF were normalized against DAPI signal.

## Senescence induction

Cells were treated with recombinant IFN-β every day for 10 days. For oncogene-induced senescence (OIS), MEF and BJ cells were transduced with a retroviral vector containing pLNCX2-ER:RAS (Addgene, #67844). Transduced cells were treated with 200 nM tamoxifen every day until analysis. To induce senescence using reactive-oxygen species, MEF cells were treated with 500 μM $H_2O_2$ in a serum-free medium for 1 h. Medium was then replaced with complete media and cells were incubated for 7 days. For doxorubicin-induced senescence, MDA-MB-231 and WI-38 cells were treated with 50 and 250 nM doxorubicin, respectively, for 24 h. Medium was then replaced, and cells were further incubated 7 days. For ABT-induced senescence, BJ, WI-38 and MEF cells were treated with 5 μM ABT-737 (Selleckchem). Every 48 h, media was replaced with fresh media containing ABT, and treatment was repeated over a period of 21 days (for MEF, 7 days). MDA-MB-213 cells were treated with 10 μM ABT-737 for 24 h. Media was then replaced and cells were further incubated for 7 days in normal media.

## Generation of conditioned medium

HaCaT-ICAD-mAID-GFP cells were treated with auxin as described in Fig. 4A. After incubation, medium was then collected, centrifuged 5 min at 2000 × g and sterile filtered. The conditioned medium was mixed 1:1 with complete normal media and used for experiments.

## ELISA

Supernatants were centrifuged at 3000 × g. ELISA was performed following the instructions of the mouse IL-6 ELISA Ready-SET-Go! (eBioscience, #88-7064-88) or the human IL-6/IL-8 ELISA MAX Deluxe (both from BioLegend, #430516, #431504). For LEGENDplex, LEGENDplex Mouse Anti-Virus Response Panel (13-plex) (740622, Biolegend) was used, according to the manufacturer's protocol.

## Click-iT EdU incorporation assay

Proliferation analysis was performed using the Click-iT EdU Imaging kit as instructed by the manufacturer (Fisher

Scientific, C10340). Tile scans (5 × 5) were taken at random with a confocal microscope, and images were analyzed using ImageJ.

## Quantitative real-time PCR

Cells were lysed in TRI Reagent, and RNA was extracted using the Direct-zol RNA Miniprep kit (Zymo Research, R2052) according to the manufacturer's instructions. Organs were harvested, homogenized, and incubated with a proteinase K solution for 30 min before RNA extraction. In total, 500 ng of total RNA were reverse transcribed using the Revertaid First Strand cDNA synthesis kit (Thermo Fisher Scientific, K1622) and analyzed by real-time qPCR using SYBR Green and a QuantStudio 5 Real-Time PCR (Applied Biosystems). Primers used in this study are listed in Appendix Table S1.

## Statistical analyses

Statistical analyses were performed using Prism 8 (GraphPad) software. Unpaired parametric $t$ test (with Welch's correction) was used for normally distributed datasets. Data were considered significant when $P \leq 0.05$, with *$P \leq 0.05$, **$P \leq 0.01$, ***$P \leq 0.001$ or ****$P \leq 0.0001$.

# Data availability

This study includes no data deposited in external repositories.

The source data of this paper are collected in the following database record: biostudies:S-SCDT-10_1038-S44318-024-00163-9.

# Peer review information

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

## Acknowledgements

The authors thank Prof. Clemens Schmitt for the WI-38 cells. This work was supported by the Deutsche Krebshilfe (grant to G.H. 70114270). We would like to acknowledge the Lighthouse Core Facility for their assistance with confocal microscopy.

## Author contributions

**Aladin Haimovici**: Conceptualization; Data curation; Formal analysis; Supervision; Validation; Investigation; Visualization; Methodology; Writing—original draft; Writing—review and editing. **Valentin Rupp**: Investigation; Methodology. **Tarek Amer**: Investigation; Methodology. **Abdul Moeed**: Investigation. **Arnim Weber**: Investigation; Methodology. **Georg Häcker**: Conceptualization; Resources; Supervision; Funding acquisition; Validation; Writing—original draft; Project administration; Writing—review and editing.

Source data underlying figure panels in this paper may have individual authorship assigned. Where available, figure panel/source data authorship is listed in the following database record: biostudies:S-SCDT-10_1038-S44318-024-00163-9.

## Funding

## Disclosure and competing interests statement

The authors declare no competing interests.

# Expanded View Figures

**A**

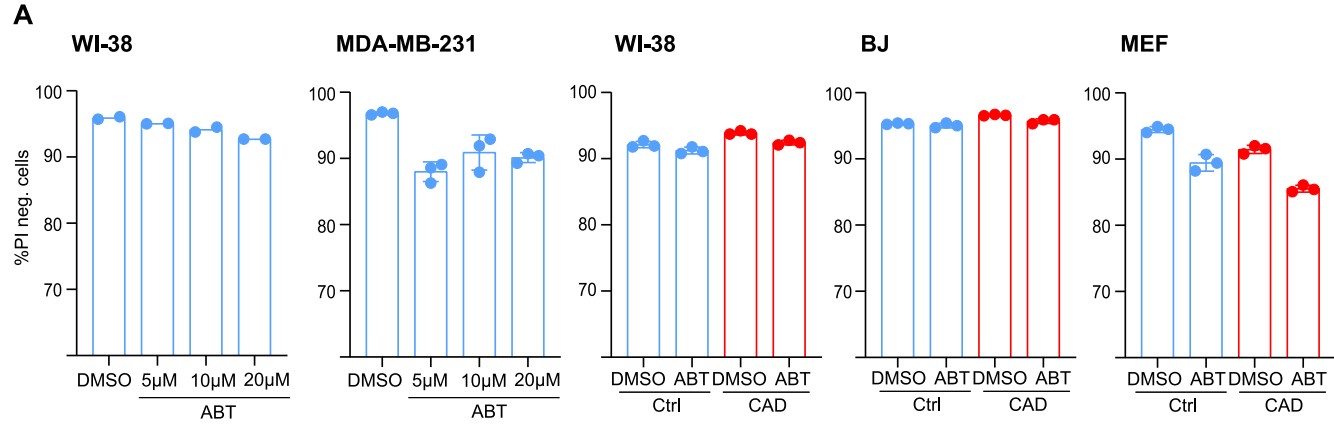

**B**

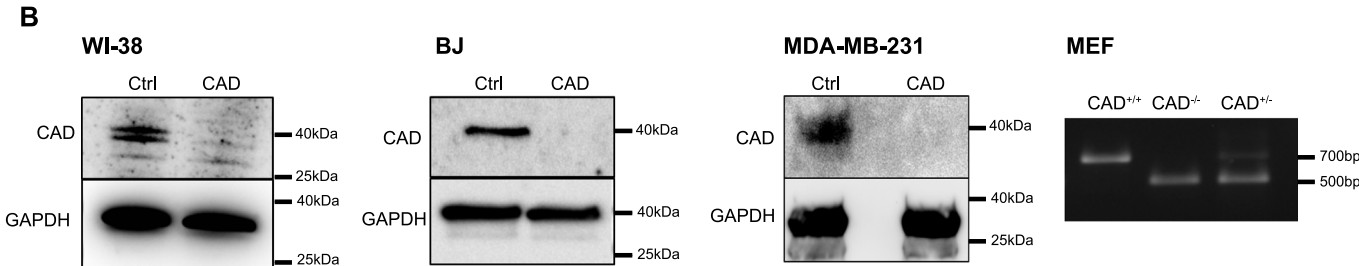

**C**

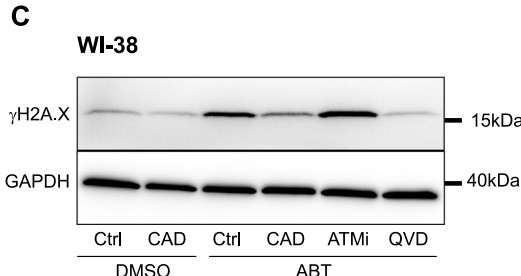

**Figure EV1. Sub-lethal CAD-activation.**

(A) Percentage of viable cells after ABT treatment (48 h) were measured by flow cytometry (PI staining). For the experiments where CAD-deficient cells are used, cells were treated with 5 µM ABT. Each symbol shows the result from one independent experiment. Data represent the mean/SD. (B) Confirmation of CAD-deletion was measured by western blot. GAPDH is used as loading control. For the MEF, genotyping for CAD is shown. (C) WI-38 cells were treated with 5 µM ABT-737 for 48 h, in the presence or not of QVD (20 µM) or the ATM inhibitor Ku55933 (10 µM). γH2A.X protein expression was analyzed by western blot. GAPDH is used as loading control. Blot is representative of 3 independent experiments. Source data are available online for this figure.

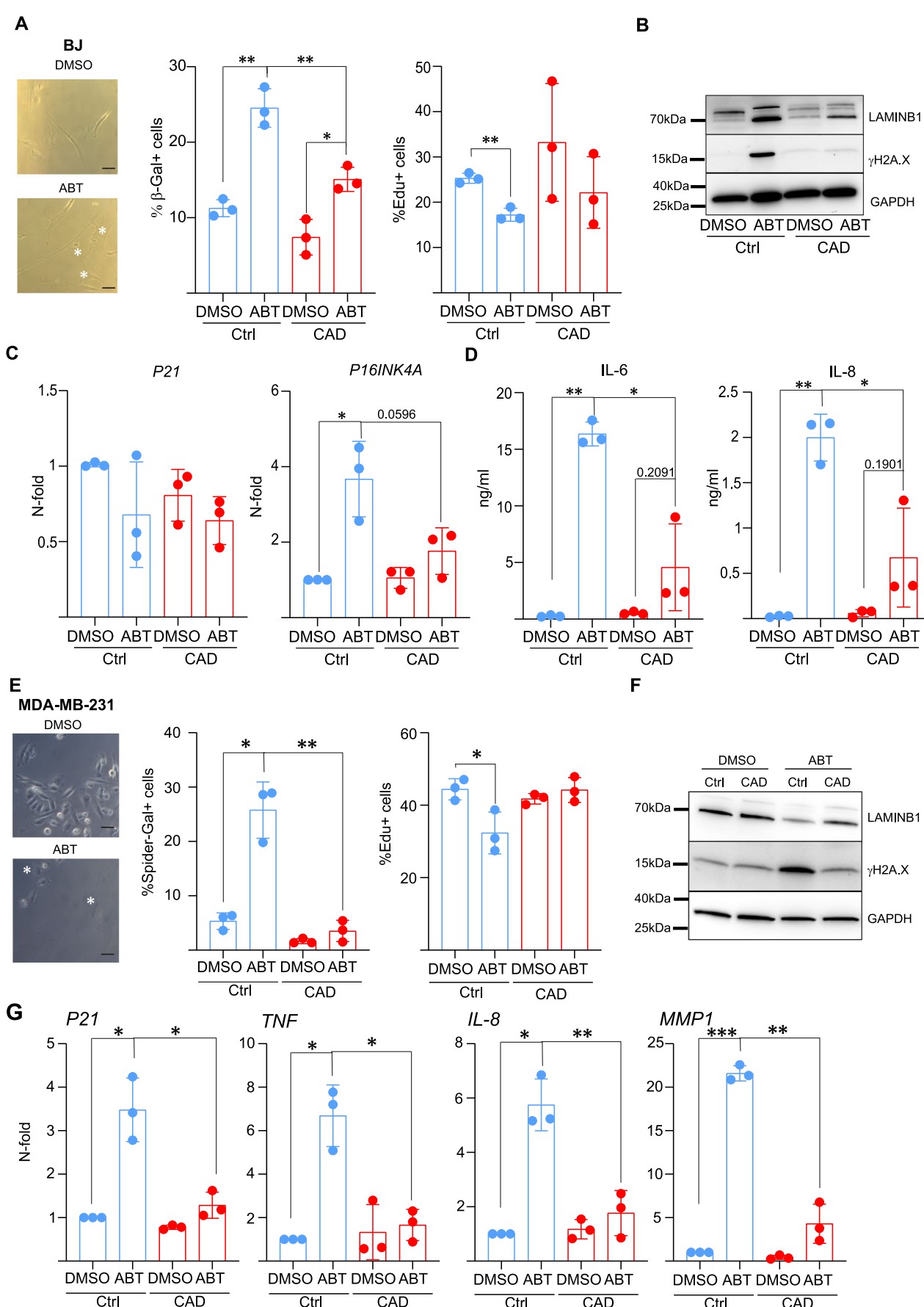

◄ **Figure EV2. Sub-lethal activation of CAD can trigger senescence.**

(A) BJ fibroblasts (carrying a non-targeting gRNA (Ctrl) or CAD-deficient) were treated with 5 μM ABT-737. Media was replaced with fresh ABT every 48 h over a period of 21 days. Bright-field images show cells after 21 days of ABT treatment. White stars highlight cells with enlarge and flat morphology. Scale bar: 50 μm. Cells were stained with β-galactosidase staining solution and Edu and percentages of SA-β-Gal$^+$ and Edu$^+$ cells were quantified by microscopy. (B) Cells were treated as in (A). Lamin B1 and γH2A.X protein expression was analyzed by western blot. GAPDH was used as loading control. (C) Expression of senescence-associated genes was measured by RT-PCR. (D) Supernatants from cells cultured for 21 days were analyzed by ELISA for IL-6 and IL-8. (E) MDA-MB-231 cells (carrying a non-targeting gRNA (Ctrl) or CAD-deficient) were treated with 10 μM ABT-737 for 24 h, washed and incubated for 7 days in normal media. Bright-field images show cells after 7 days of ABT treatment. Scale bar: 50 μm. White stars highlight cells with enlarge and flat morphology. SA-β-Gal activity was measured using Spider-Gal. Proliferation was measured with Edu incorporation. Percent positive cells are shown. (F) Lamin B1 and γH2A.X protein expression was analyzed by western blot. GAPDH was used as loading control. (G) Expression of senescence-associated genes was measured by RT-PCR. Each symbol shows the result from one independent experiment. Data represent the mean/SD. Unpaired parametric $t$ test (with Welch's correction) was used to calculate statistical significance. *$P < 0.05$, **$P < 0.01$, ***$P < 0.001$. Source data are available online for this figure.

                                                                

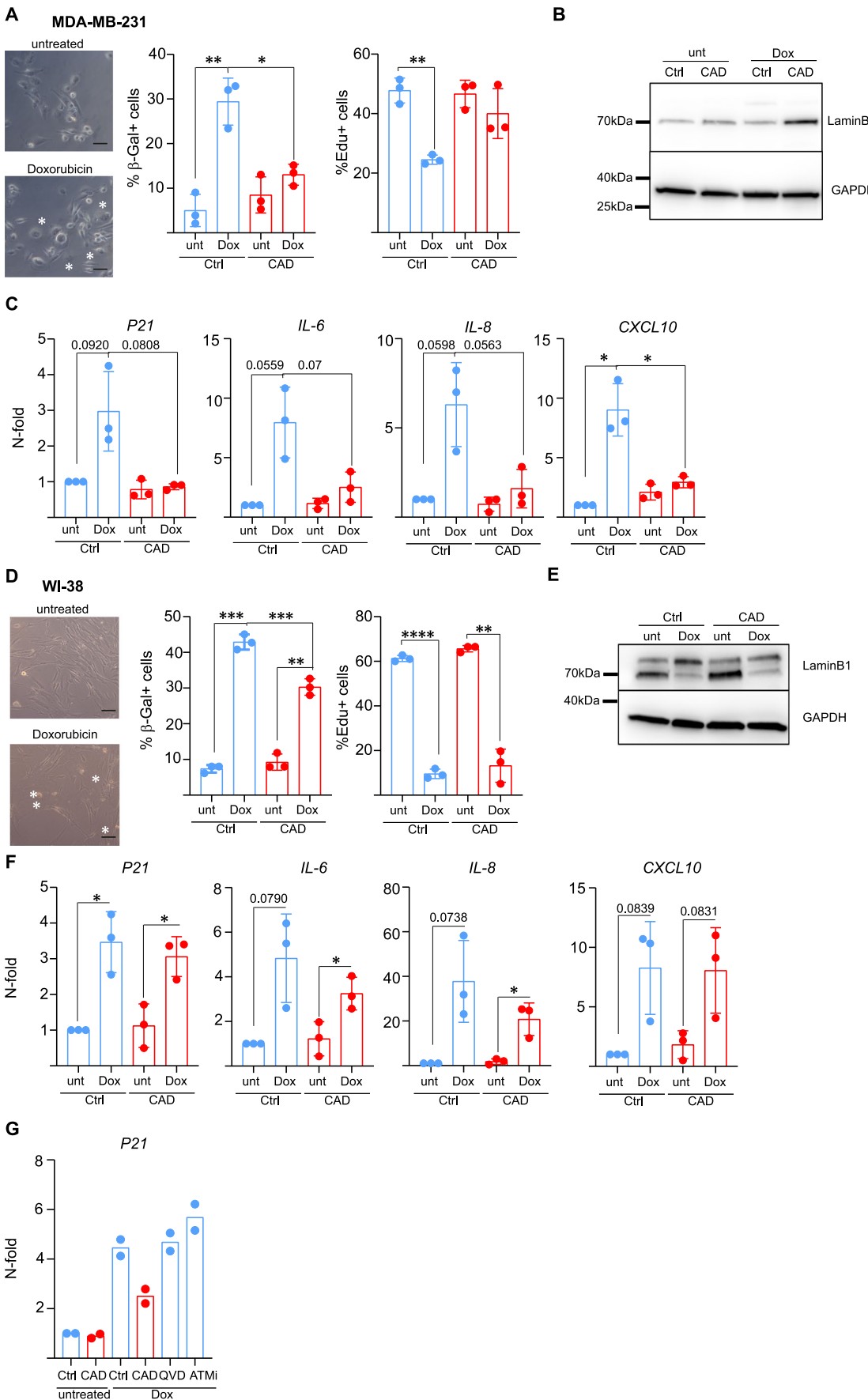

◀  **Figure EV3.  CAD-activation may contribute to doxorubicin-induced senescence.**

(A) MDA-MB-231 cells (carrying a non-targeting gRNA (Ctrl) or CAD-deficient) were treated with 50 nM Doxorubicin for 24 h. Medium was replaced and cells were incubated for 7 days. Bright-field images show cells after 7 days of Doxorubicin treatment. White stars highlight cells with enlarge and flat morphology. Scale bar: 50 μm. Cells were stained with β-galactosidase and Edu staining solution and percentages of SA-β-Gal⁺ and Edu⁺ cells were quantified by microscopy. (B) Lamin B1 protein expression was analyzed by western blot. GAPDH was used as loading control. (C) Expression of senescence-associated genes was measured by RT-PCR. (D) WI-38 cells (carrying a non-targeting gRNA (Ctrl) or CAD-deficient) were treated with 250 nM Doxorubicin for 24 h. Medium was replaced and cells were incubated for 7 days. Bright-field images show cells after 7 days of Doxorubicin treatment. White stars highlight cells with enlarge and flat morphology. Scale bar: 50 μm. Cells were stained with β-galactosidase and Edu staining solution and percentages of SA-β-Gal⁺ and Edu⁺ cells were quantified by microscopy. (E) Lamin B1 protein expression was analyzed by western blot. GAPDH was used as loading control. (F) Expression of senescence-associated genes was measured by RT-PCR. (G) MDA-MB-231 cells were treated as in (A), in presence or absence of QVD or ATM inhibitor. P21 expression was measured by RT-PCR after 7 days of treatment. Each symbol shows the result from one independent experiment. Data represent the mean/SD. Unpaired parametric *t* test (with Welch's correction) was used to calculate statistical significance. *P < 0.05, **P < 0.01, ***P < 0.001, ****P < 0.0001. Source data are available online for this figure.

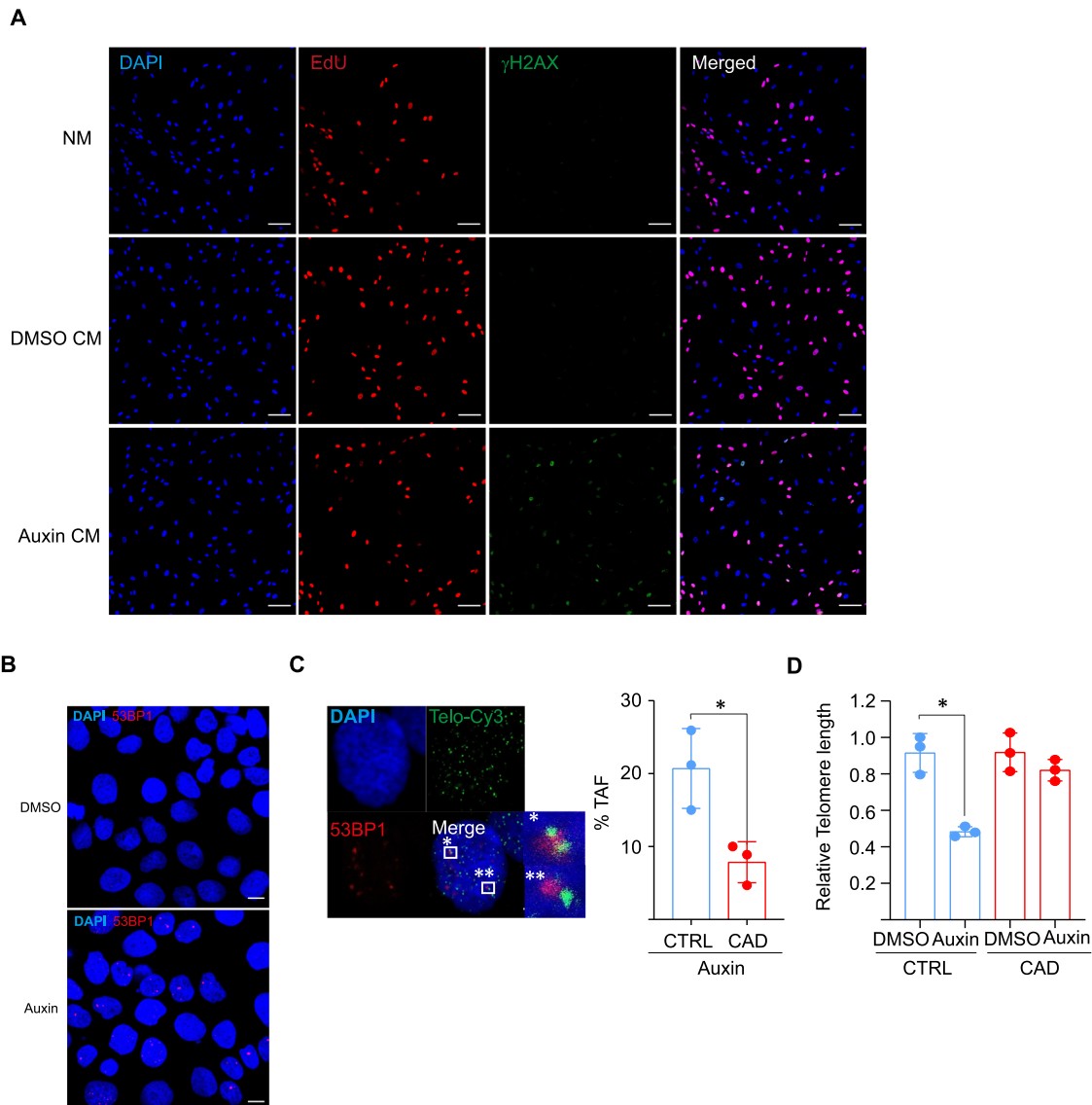

**Figure EV4. A CAD-induced DDR at telomeres.**

(A) Representative fluorescent images of BJ fibroblasts after incubation with normal media (NM), DMSO or Auxin condition media. Cells were stained for Edu, γH2A.X and DAPI. Scale bar: 10 μm. (B) HaCaT-ICAD-mAID-GFP cells were treated with auxin for 6 h, fixed and stained for 53BP1 (DDR) with DAPI (DNA). Scale bar: 10 μm. (C) HaCaT-ICAD-mAID-GFP cells were treated with auxin for 6 h, fixed and stained for 53BP1 (DDR), for telomeres (using a telomere-specific DNA-probe) and with DAPI (DNA). Images are maximum intensity projections of at least 20 planes. Amplified images on the right (* and **) are from single Z planes where colocalization (*)/close proximity (**) was found. Frequencies of TAF in HaCaT-ICAD-mAID-GFP cells and control cells (HaCaT cells expressing only the Tir1 F-box protein but not the degradable ICAD; see "Methods") upon auxin treatment for 6 h is shown. Results are expressed as percentage of 53PB1 foci that co-localized with the telomere probe. Symbols show results from independent experiments. Data represent the mean/SD. (D) Relative telomere length in HaCaT-ICAD-mAID-GFP and HaCaT-ICAD-mAID-GFP CAD-deficient treated with DMSO or auxin for 6 h. Symbols show results from independent experiments ($n = 3$). Data represent the mean/SD. Unpaired parametric $t$ test (with Welch's correction) was used to calculate statistical significance. *$P < 0.05$. Source data are available online for this figure.

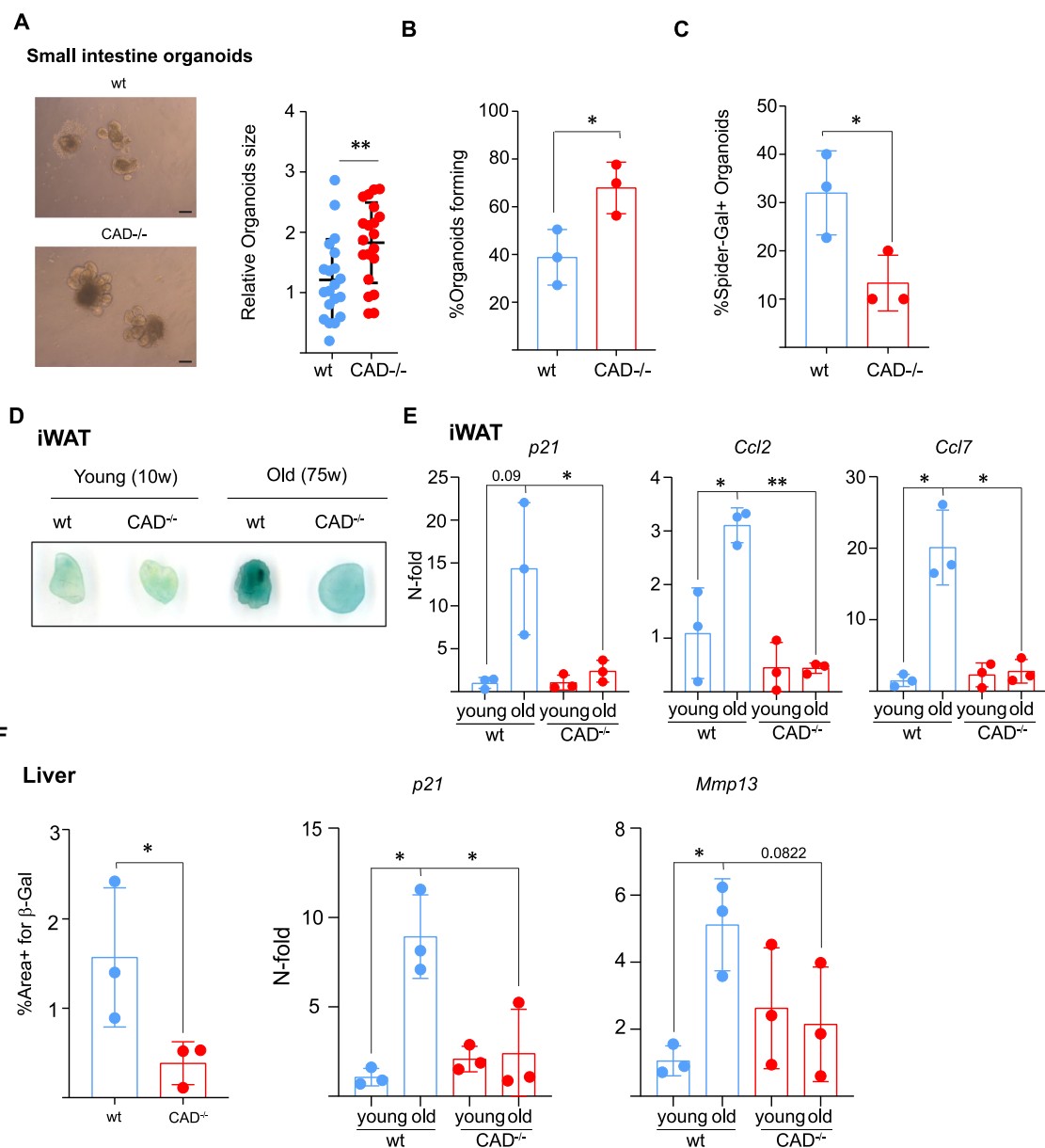

**Figure EV5. Small intestinal organoids, adipose tissue and liver were isolated from old (75 weeks) wt and CAD-deficient mice.**

(A) Exemplary organoids (left) and quantification of organoid size (right) after 5 days in culture are shown. Scale bar: 10 μm. Each symbol shows the size of one organoid. Twenty organoids per group were measured. Data represent the mean/SD. (B) Fifty intestinal crypts were seeded per intestine and the percentage of organoids forming after 5 days were quantified. Each symbol represents one mouse/intestine. Data represent the mean/SD. (C) Organoids were subjected to staining for SA-β-Gal using Spider-gal. Percentage of organoids staining positive for each mouse are given. Each symbol represents one mouse/intestine ($n = 3$ mice per group). Data represent the mean/SD. Unpaired parametric t test (with Welch's correction) was used to calculate statistical significance. *$P < 0.05$, **$P < 0.01$. (D) Inguinal white adipose tissue (iWAT) was isolated from three young and three old mice of each genotype. Tissues were stained for SA-β-Gal activity. Image is representative of 4 stained WAT tissues per group. (E) Expression of senescence-associated genes from iWAT samples. Gene expression was measured by RT-PCR. Young (8–10 weeks) ($n = 3$) and old (75 weeks) ($n = 3$) mice per genotype were analyzed. Data represent the mean/SD. (F) Liver cryosections from 75-week-old mice were stained for SA-β-Gal activity and the positive area was measured by microscopy (left panel). Each symbol represents one mouse. Expression of senescence-associated genes in liver samples (right panel). Gene expression was measured by RT-PCR. Young (8–10 weeks) ($n = 3$) and old (75 weeks) ($n = 3$) animals were analyzed. Each symbol represents one mouse. Data represent the mean/SD. Unpaired parametric t test (with Welch's correction) was used to calculate statistical significance. *$P < 0.05$. Source data are available online for this figure.

