## [Peer Review File · The EMBO Journal]

The caspase-activated DNase promotes cellular senescence

Aladin Haimovici, Valentin Rupp, Tarek Amer, Abdul Moeed, Arnim Weber, and Georg Häcker

Corresponding author(s): Georg Häcker (georg.haecker@uniklinik-freiburg.de) , Aladin Haimovici (aladin.haimovici@uniklinik-freiburg.de)

Review Timeline:

Submission Date:	29th Sep 23
Editorial Decision:	28th Nov 23
Revision Received:	30th Apr 24
Editorial Decision:	6th Jun 24
Revision Received:	10th Jun 24
Editorial Decision:	14th Jun 24
Revision Received:	15th Jun 24
Accepted:	19th Jun 24

Editor: Ioannis Papaioannou

Transaction Report:

Dear Prof. Häcker,

Thank you for submitting your manuscript for consideration by The EMBO Journal. It has been seen by three experts in the field, and we have received the complete set of their comments, which I have already shared with you (included again below). I would like to thank you for your point-by-point response to them and your provisional revision plan.

Referees #1 and #3 find the work important and interesting, although they also identify a number of limitations and raise several concerns, which should be addressed. Several assays and analyses should be repeated using additional cell types and primary cells, as suggested by the referees and along the lines of your revision plan. I would like to note that investigation of the ABT-737 effects in additional cell types -including the suggested experiment using procured WI-38 fibroblasts- is highly recommended. Referee #2 is more critical and emphasizes that: i. the used methodology for detecting senescence is insufficient for supporting the conclusions of the study; and ii. no mechanistic insight is provided into how the Caspase-Activated DNase induces senescence. For further consideration of the manuscript at The EMBO Journal, all technical concerns of the referee, including the need to use additional markers and controls in some assays, should be addressed. Regarding the lack of mechanistic insight, we do agree with the reviewer, but considering the significance of the finding, we will consider the manuscript further if some -but not necessarily complete- mechanistic insight is provided in a revision version of the manuscript, according to the suggestions you make in your revision plan.

Given the referees' comments and recommendations, as well as your willingness to address the raised concerns, I would like to invite you to submit a revised version of your manuscript, addressing the comments of all three reviewers. I should add that it is EMBO Journal policy to allow only a single round of major revision, and acceptance of your manuscript will therefore depend on the completeness of your responses in this revised version. If you have any questions or comments, we can also discuss the revisions in a video chat, if you like.

We generally allow three months as standard revision time (27th February 2024). As a matter of policy, competing manuscripts published during this period will not negatively impact our assessment of the conceptual advance presented by your study. However, we request that you contact us as soon as possible upon publication of any related work, to discuss how to proceed. Should you foresee a problem in meeting this three-month deadline, please let us know in advance and we may be able to grant an extension.

Thank you for the opportunity to consider your work for publication in The EMBO Journal. I look forward to your revision.

Yours sincerely,

Instructions for preparing your revised manuscript

1. When you are ready to submit the revision, please upload:

- A Word file of the manuscript text (including legends of main Figures, EV Figures and Tables). Please make sure that changes are highlighted (or "tracked") to be clearly visible.

- Individual production-quality figure files (one file per figure). When assembling your figures, please refer to our figure preparation guidelines in order to ensure proper formatting and readability in print as well as on screen:

If the data shown in a figure are obtained from n {less than or equal to} 2, please use scatter plots showing the individual data points.

- i. the name of the statistical test used to generate error bars and P values
- ii. the number (n) of independent experiments (please specify technical or biological replicates) underlying each data point (discussion of statistical methodology can be reported in the Materials and Methods section, but figure legends should contain a basic description of n , P , and the test applied)
- iii. the nature of the bars and error bars (s.d., s.e.m.).

- A point-by-point response to the referees' comments, with a detailed description of the changes made (as a word file). All referees' concerns must be fully addressed and their suggestions taken on board. When preparing your letter of response to the referees' comments, please bear in mind that this will form part of the Review Process File and will therefore be available online to the community. Please note that you have the possibility to opt out of the transparent process at any stage prior to publication by letting the editorial office know (contact@embojournal.org); if you do opt out, the Review Process File link will point to the following statement: "No Review Process File is available with this article, as the authors have chosen not to make the review process public in this case.". For more details on our Transparent Editorial Process, please visit our website: <https://www.embopress.org/page/journal/14602075/authorguide#transparentprocess>

- Expanded View (EV) files (replacing Supplementary Information) that are collapsible/expandable online. A maximum of 5 EV Figures can be typeset. EV Figures should be cited as "Figure EV1, Figure EV2" etc. in the text, and their respective legends should be included in the manuscript file after the legends of regular figures. See detailed instructions regarding Expanded View files here: <https://www.embopress.org/page/journal/14602075/authorguide#expandedview>

- For the figures that you do NOT wish to display as Expanded View figures, they should be bundled together with their legends in a single PDF file called "Appendix", which should start with a short Table of Contents (including page numbers). Appendix figures should be referred to in the main text as: "Appendix Figure S1, Appendix Figure S2" etc. Please see detailed instructions here: <https://www.embopress.org/page/journal/14602075/authorguide#expandedview>

- A complete author checklist, which you can download from our author guidelines (<https://www.embopress.org/page/journal/14602075/authorguide>). Please note that the checklist will also be part of the Review Process File.

2. Please note that no statistics should be calculated if $n=2$.

3. Before submitting your revision, primary datasets (and computer code, where appropriate) produced in this study need to be deposited in appropriate public databases (see <https://www.embopress.org/page/journal/14602075/authorguide#dataavailability>).

The accession numbers and database should be listed in a formal "Data availability" section (placed after Materials and Methods) that follows the model below (see also <https://www.embopress.org/page/journal/14602075/authorguide#dataavailability>):

Data availability

- RNA-seq data: Gene Expression Omnibus GSE46843 (<https://www.ncbi.nlm.nih.gov/geo/query/acc.cgi?acc=GSE46843>)
- [data type]: [name of the resource] [accession number/identifier/doi] ([URL or identifiers.org/DATABASE:ACCESSION])

*** Note: all links should resolve to a page where the data can be accessed. ***

*** Note: the Data Availability Section is restricted to new primary data that are part of this study. ***

*** Note: In case you have no data that require deposition in a public database, please state so instead of refereeing to the database: "Our study includes no data deposited in public repositories." under the heading "Data availability". ***

4. Please check that the title and the abstract of the manuscript are brief, yet explicit, even to non-specialists. The length of the title should not exceed 100 characters, and the abstract should be a single paragraph not exceeding 175 words.

5. Please also note our reference format: <https://www.embopress.org/page/journal/14602075/authorguide#referencesformat>.

7. Please remember: digital image enhancement is acceptable practice, as long as it accurately represents the original data and conforms to community standards. If a figure has been subjected to significant electronic manipulation, this must be noted in the figure legend or in the "Materials and Methods" section. The editors reserve the right to request original versions of figures and the original images that were used to assemble the figure.

8. Our journal encourages inclusion of data citations in the reference list to directly cite datasets that were obtained from public databases. Data citations in the article text are distinct from normal bibliographical citations and should directly link to the database records from which the data can be accessed. In the main text, data citations are formatted as follows: "Data ref: Smith et al, 2001" or "Data ref: NCBI Sequence Read Archive PRJNA342805, 2017". In the Reference list, data citations must be labeled with "[DATASET]". A data reference must provide the database name, accession number/identifiers, and a resolvable link to the landing page from which the data can be accessed at the end of the reference. Further instructions are available at: <https://www.embopress.org/page/journal/14602075/authorguide#referencesformat>.

9. We request authors to consider both actual and perceived competing interests. Please review our policy (<https://www.embopress.org/page/journal/14602075/authorguide#conflictsinterest>) and update your competing interests statement if necessary. Please name this section 'Disclosure and competing interests statement' and place it after the Acknowledgements section.

10. Please note that all corresponding authors are required to provide an ORCID ID upon submission of a revised manuscript (<https://orcid.org/>). Please find instructions on how to link your ORCID ID to your account in our manuscript tracking system in our Author guidelines (<https://www.embopress.org/page/journal/14602075/authorguide#authorshipguidelines>).

11. We use CRediT to specify the contributions of each author in the journal submission system. CRediT replaces the author contribution section, which should be removed from the manuscript. Please use the free text box to provide more detailed descriptions. See also guide to authors: <https://www.embopress.org/page/journal/14602075/authorguide#authorshipguidelines>.

13. We would also welcome the submission of cover suggestions or motifs to be used by our Graphics Illustrator in designing a cover.

14. Please use the link below to submit your revision:
<https://emboj.msubmit.net/cgi-bin/main.plex>

Referee #1:

This study investigates the role of Caspase-Activated DNase (CAD) in cellular senescence, which is characterized by permanent cell cycle arrest and the secretion of senescence-associated factors. While CAD is traditionally associated with apoptosis, this research highlights its broader involvement in senescence processes. The study demonstrates CAD's significance in various scenarios that induce senescence; these include oncogene-induced senescence, interferon-induced senescence, and the effects of chemotherapeutic drugs. CAD activation is shown to be a pivotal mediator in these processes, emphasizing its wide-ranging involvement in senescence. Furthermore, the research shows that CAD's activation alone can induce senescence. Lastly, the research establishes CAD's role in age-associated cellular senescence in mice, indicating its influence on aging-related conditions. In conclusion, this research uncovers CAD as a key player in cellular senescence and aging processes, with potential implications for therapeutic strategies targeting senescence-related diseases. This study is a complement to the recent study by Larsen et al. (2023), *Science*. Although the findings are intriguing and important, I have several concerns, and I would not recommend the manuscript for publication without addressing those concerns.

1. The hypothesis depicted in Figure 1 suggests that sub-lethal CAD activation may induce senescence. However, using the MDA-MD231 breast cancer cell line in this study could be misleading, especially considering that p53 is one of the 557 mutated genes in MDA-MB-231, and p53 is pivotal in senescence through DNA damage repair (DDR) pathways and p21 activation. Therefore, additional investigations are imperative to determine if sub-lethal CAD activation induces senescence in commonly employed primary cells for senescence model, such as IMR-90, WI-38, BJ cells, and MEFs.
2. However, in a study by Song et al. (2011) in *Cancer Research*, it was observed that the WI-38 cell line exhibited resistance to ABT-737-induced senescence. Therefore, claiming that sub-lethal CAD activation universally triggers senescence may be an overstatement.
3. It is essential to determine how the authors confirmed that the ABT-737 treatment activated CAD, resulting in senescence in this experiment. My preference would be to use primary cells for this confirmation.
4. In Figure 2, the shift from MEF to the cancerous cell line MDA-MD231 for DNA-damage-induced senescence experiments raises questions. This change in cell lines might imply that the authors didn't observe the desired effect in primary cells, prompting the switch. Notably, CAD-dependent breaks were detected in various human malignant cell lines but were absent in

nonmalignant cells (Larsen et al., 2023, Science). How did the authors reconcile these observations?

5. Figure 4 also exhibits a change in cell line to HaCaT, a cancerous cell line, for CAD activation. It remains unclear why the HaCaT cell line was suddenly chosen for this purpose. Considering that Figures 4 and 1 present similar results, there is potential to consolidate them into a single figure. However, there is still uncertainty regarding whether CAD activation alone is adequate to induce senescence in primary cells.

6. In the discussion section, the statement, "It was surprising to observe that doxorubicin, which intercalates in and directly damages DNA also depends on CAD for senescence induction," might be an overgeneralization. It would be beneficial to investigate whether CAD is necessary for DNA damage-induced senescence in primary cells by using gamma or X-ray irradiation and treatments with agents like etoposide or doxorubicin on primary fibroblasts.

7. CCF serves as a marker of senescence and is an integral component in the production of SASP. A study by Vizioli et al. (2020) in *Genes and Development* proposed that the Mitochondria-ROS-JNK signaling pathway drives CCFs and the SASP. Another research by Ichim et al. (2015) in *Molecular Cell* demonstrated the requirement of CARD for JNK1/2 activation and subsequent γ H2A.X induction. However, the question remains whether CARD is essential for CCF formation. The authors mentioned, "We and others have reported previously that CAD activation can generate CCF. DNA-recognition downstream of CAD activation seems a likely contributor to the generation of the SASP and the induction of senescence." This assertion is based on extrapolation without specific references to CCF. The given references did not discuss about CCF formation. The confusion might stem from the distinction between Micronuclei (MN) and CCF. These events have different characteristics and mechanisms of formation, despite sharing common molecular markers. MN formation occurs in the context of cell division and is linked to chromosome missegregation, while CCF formation is associated with the loss of nuclear membrane integrity in senescent cells (Miller et al., 2021, *Cell*).

Referee #2:

The manuscript by Haimovici et al. aims to demonstrate that the cellular endonuclease Caspase-Activated DNase (CAD) is an important factor in senescence induction. The authors claim that CAD activation by sub-lethal signals in the apoptosis pathway is responsible for the initiation of senescence. Additionally, they find that various inducers of cellular senescence, such as oncogenic RAS, interferon and doxorubicin treatment, depend on CAD in their capacity to trigger senescence. Finally, the authors tested the contribution of CAD to senescence in various mouse tissues and found reduced senescence in organs of aged CAD-deficient mice. Unfortunately, the study is too preliminary in nature and most (if not all) of the claims are devoid of robust experimental support. Importantly, there is a complete lack of mechanistic insight into how CAD may activate senescence-related factors. In general, the manuscript lacks depth and the methodology is flawed in many parts. Some specific remarks:

1. In Figures 1A-B the authors assess induction of senescence in MDA-MB-231 breast cancer cells upon ABT treatment (inactivating anti-apoptotic factors) in Control (Ctrl) and CAD-deficient cells. The authors' approach relies on estimation of SA- β -Gal levels and the expression of a few other markers. However, this approach is not sufficient any more in establishing the presence of senescence in any biological sample or cell culture. As explained in the senescence detection algorithm established by Kohli et al. (2021), SA- β -Gal or lipofuscin levels should be carefully quantified, along with expression of proliferation markers (such as Ki-67) and major senescence-related markers (p16, p21 and lamin B1). The approach followed by the authors is incomplete as they do not present changes in proliferation nor expression of key senescence markers (p16, lamin B1). This is an important omission observed throughout the manuscript (e.g. Fig. 2A-B and H-I, 3A etc), therefore it is impossible to determine whether senescence is correctly estimated in each case.

2. In Figures 1A-B, there is no data showing CAD expression in Ctrl versus CRISPR/Cas9-engineered CAD-deficient cells. This is an important control that is obviously missing. Again, in Figures 2E-G the authors do not provide any validation of CAD levels in CAD-deficient MEFs compared to controls. All these data have to be presented (for CRISPR/Cas9 experiments sequencing results have to be shown and for the mouse-derived MEFs genotyping data should be presented).

3. Many parts of the manuscript are highly speculative. For example, in page 5 the authors mention "We further exposed MEFs to H₂O₂, which directly induces DNA-damage and therefore bypasses the activation of CAD by caspases". Where do the authors base this assumption on? References and/or experiments are definitely required to support this statement.

4. The authors claim that the data in Fig. EV1 indicate the principal ability of CAD-deficient cells to undergo senescence upon DNA damage. Unfortunately, the displayed q-PCRs in Fig. EV1 are insufficient to support the claim. A series of well-designed experiments with appropriate controls with regards to senescence (see point 1 above) would be, instead, required to demonstrate the role of CAD in senescence upon DNA damage and/or oxidative stress.

5. The reliance of senescence induction on CAD upon doxorubicin treatment is not adequately investigated and the authors only provide speculations to try to interpret the results. A better explanation should be experimentally provided regarding the potential differential requirement of CAD upon H₂O₂ or doxorubicin-induced DNA damage.

6. In Figure 3B, why do the authors assess only IL-6 levels in the supernatant of MEFs in regards to SASP? Deriving a complete SASP profile would be much more informative and absolutely required to make statements regarding the SASP.

7. In Figures 4E-G the authors show that incubation of BJ fibroblasts with medium from HaCaT cells expressing active CAD leads to BJ reduced proliferative capacity (estimated by EdU incorporation), signs of a DDR and loss of lamin B1. However, showing only EdU incorporation changes (which appear also statistically insignificant) and percentages of H2AX+ cells without any representative image of relevant stainings does not convincingly indicate DDR. Surprisingly, loss of lamin B1 is never shown, but the authors rather refer to it as a fact. In general, the results are too weak to support most of the statements in the

manuscript.

8. The claim that CAD induces DDR at telomeres is already known and described, also according to the authors (ref. 29 in the manuscript). Additionally, Figure 4H has two serious problems: no control is shown and only one nucleus is presented. No conclusions can be safely drawn, but there seems to be little point in doing those experiments as the findings are no longer novel. The potential novelty of the study is somewhat compromised by previous findings.

9. For in vivo data, senescence validation experiments need to be appropriately repeated (see point 1 above). In Figure 5E, extremely high levels of senescence are displayed in wt tissue, which does not appear reasonable as the intestine is known for displaying high renewal rates, and may be attributed to non-specific SA- β -Gal absorption by the mucous. In any case, the displayed results are not sufficient to strongly support the claims put forward by the authors. A much more thorough validation with updated senescence detection tools and assays is required.

10. In their Discussion (page 10), the authors mention an interesting point, on which their whole manuscript is based: that activation of CAD within a certain signaling threshold may lead to senescence, otherwise other processes such as apoptosis may be induced. However, this notion is not experimentally tested at all and remains purely theoretical. Mechanistic insights to justify a potential interplay between CAD and known molecular determinants of senescence are completely absent.

Referee #3:

Based on description of sub-lethal caspase activity leading to CAD (DNase) activation made by this lab and others, Haimovici and colleagues investigated a role for CAD in senescence. Summarizing the paper, the authors demonstrate a convincing role for CAD following different senescence triggers (DNA-damage, OIS, replicative) and demonstrate that CAD activation is itself sufficient to cause senescence. This exciting work makes an important contribution to emerging data linking apoptotic signaling and senescence. The data largely support the authors' conclusions. That said, I have some points that should be addressed.

- trying to get a sense of the absolute requirement of CAD in senescence (vs contributory factor), in the context of OIS, dox and replicative senescence shown here, do the CAD deficient cells eventually undergo senescence to the same degree as WT cells or is there always a persistent reduction (in as far as the authors have analyzed)?

- as the authors note the role of CAD in doxorubicin mediated senescence is somewhat surprising given that DOX is a potent DNA-damage inducer, as discussed, this raises the possibility that CAD activation independent of caspase activity following DNA-damage (as described in a recent Science paper) is mediating senescence in this context. This should be directly tested, one way would be to overexpress non-caspase cleavable ICAD in these cells, asking do they undergo senescence to the same degree - if so, demonstrates non-caspase dependent activation of CAD in the context of doxorubicin mediated senescence, which is an important message.

- given the evident negative role of senescence burden impacting longevity and/or health, can the authors comment on the phenotype of aged CAD deficient mice, for instance, do they display any evident extension of lifespan and/or health?

- in terms of analysis of senescence, it would be helpful if images of cells in Figure 1 and 2 (ABT-737, OIS and dox) are included and commented on, do the cells display typical senescent morphology? Secondly, do the B gal positive cells display cell cycle arrest?

- for the CRISPR lines, blots demonstrating CAD deletion should be included.

- it would be helpful if the authors could discuss their findings in the context of other recent publications linking apoptotic signaling and senescence, please see PMID 36414711, 37821702.

Point-by-point-reply

We would like to thank the reviewers for their thorough analysis and comments on our manuscript. We have addressed the points raised and will comment on them individually below. There is no doubt that the manuscript has much improved in this process.

Referee #1

This study investigates the role of Caspase-Activated DNase (CAD) in cellular senescence, which is characterized by permanent cell cycle arrest and the secretion of senescence-associated factors. While CAD is traditionally associated with apoptosis, this research highlights its broader involvement in senescence processes. The study demonstrates CAD's significance in various scenarios that induce senescence; these include oncogene-induced senescence, interferon-induced senescence, and the effects of chemotherapeutic drugs. CAD activation is shown to be a pivotal mediator in these processes, emphasizing its wide-ranging involvement in senescence. Furthermore, the research shows that CAD's activation alone can induce senescence. Lastly, the research establishes CAD's role in age-associated cellular senescence in mice, indicating its influence on aging-related conditions. In conclusion, this research uncovers CAD as a key player in cellular senescence and aging processes, with potential implications for therapeutic strategies targeting senescence-related diseases. This study is a complement to the recent study by Larsen et al. (2023), Science. Although the findings are intriguing and important, I have several concerns, and I would not recommend the manuscript for publication without addressing those concerns.

Thank you for the positive comments and the thoughts. We have done the experiments suggested in the comments below.

1. The hypothesis depicted in Figure 1 suggests that sub-lethal CAD activation may induce senescence. However, using the MDA-MD231 breast cancer cell line in this study could be misleading, especially considering that p53 is one of the 557 mutated genes in MDA-MB-231, and p53 is pivotal in senescence through DNA damage repair (DDR) pathways and p21 activation. Therefore, additional investigations are imperative to determine if sub-lethal CAD activation induces senescence in commonly employed primary cells for senescence model, such as IMR-90, WI-38, BJ cells, and MEFs.

We fully agree that the use of cancer cells is not optimal. Obviously, we had relied on the results from the mice to illustrate the role of CAD in non-transformed cells but it is, as stated by the reviewer, important to use other cells *in vitro* also. We have, as suggested, used these fibroblasts (WI-38, BJ and MEFs; in the human lines we deleted CAD by CRISPR, MEFs were from gene-deficient mice). We used the ABT-737 protocol, which is the best characterized stimulus; as shown elsewhere in the manuscript, CAD-activation by ABT-737 requires caspases (and the next point by the reviewer specifically asks about this stimulus). We analysed cell morphology, β -Gal-positivity, proliferation (Edu), loss of lamin B, γ H2AX and mRNA-expression for a number of senescence-associated genes. All assays showed the requirement for CAD in the senescence-associated changes. The data are shown in Fig. 1 (WI-38, MEF) and Fig. EV2 (BJ and the originally shown MDA-MB-231 cells).

2. However, in a study by Song et al. (2011) in Cancer Research, it was observed that the WI-38 cell line exhibited resistance to ABT-737-induced senescence. Therefore, claiming that sub-lethal CAD activation universally triggers senescence may be an overstatement.

ABT-737 is a BCL-2 (and less efficiently BCL-XL) inhibitor, which activates caspases corresponding to the dependency of a cell on BCL-2. Different cells can have quite differing sensitivities to ABT-737, and it is unsurprising that one condition that works for one cell does not work for the next cell. This is certainly the case for apoptosis induction, and very likely also applies to senescence induction (which uses the same pathway). This suggests that the failure in this study to induce senescence in WI-38 cells by ABT-737 may not be due to CAD not being required but to CAD not being activated in the first place.

The Song-study investigated only one condition (10 μ M ABT-737 without titration, single treatment). There would be a number of ways of how to change the conditions, such as adding an MCL-1-inhibitor. To stay with ABT-737, we chose to prolong the treatment by

replacing the medium with fresh ABT-737 every 48 h (ABT-737 at 5 μ M), followed by analysis after 21 days. As already said, Fig. 1 shows CAD-dependent induction of senescence by ABT-737 in WI-38 by all parameters tested. The results therefore suggest that in the Song-protocol ABT-737-treatment was insufficient for the induction of sub-lethal signals to the required degree.

3. It is essential to determine how the authors confirmed that the ABT-737 treatment activated CAD, resulting in senescence in this experiment. My preference would be to use primary cells for this confirmation.

We have added for all primary cells (WI-38, BJ, MEF) blots showing a CAD-dependent DNA-damage response (Fig. 1, Fig. EV2). We have also added a short-term experiment showing that ABT-737 causes a CAD-dependent γ H2AX-signal (WI-38 cells). The signal was prevented by adding the caspase-inhibitor QVD-OPH but not by ATM-inhibition (Fig. EV1C) (ATM has also been shown to be able to activate CAD as quoted by the reviewer (PMID: 35482866)). The results show that this indeed occurs through sub-lethally activated caspases (there is almost no cell death induced by this treatment, Fig. EV1B).

4. In Figure 2, the shift from MEF to the cancerous cell line MDA-MD231 for DNA-damage-induced senescence experiments raises questions. This change in cell lines might imply that the authors didn't observe the desired effect in primary cells, prompting the switch. Notably, CAD-dependent breaks were detected in various human malignant cell lines but were absent in nonmalignant cells (Larsen et al., 2023, Science). How did the authors reconcile these observations?

We can see that it may seem that we changed cells because we failed to observe an effect. This was however not the case; we simply started with different cellular models that were based on literature reports with the various stimuli. Please note that the Larsen paper did not show that CAD does not cause DNA-breaks in non-malignant cells. Those authors showed rather that the downstream effects of CAD-activity on proliferation are not found in non-malignant cells (they used RPE cells as the sole non-cancerous type). Indeed, CAD cleaves DNA in all tested cells during apoptosis, and there seems no reason to assume it would not do that in cells during sub-lethal signalling. However, the question is certainly justified. We had in an earlier report shown CAD-dependent DNA-damage (or DNA-damage response) during viral infection in MEFs (PMID: 30979778). As described above, we have now added the human fibroblasts, which we hope clarified this issue.

5. Figure 4 also exhibits a change in cell line to HaCaT, a cancerous cell line, for CAD activation. It remains unclear why the HaCaT cell line was suddenly chosen for this purpose. Considering that Figures 4 and 1 present similar results, there is potential to consolidate them into a single figure. However, there is still uncertainty regarding whether CAD activation alone is adequate to induce senescence in primary cells.

We had made Fig. 4 a separate figure because it shows that CAD is not only necessary but – in HaCaT cells – also sufficient to induce senescence. We had introduced the system into HaCaT because it requires a number of genetic manipulations (deletion of endogenous ICAD, expression of the two elements of TIR1 and the AID-ICAD-GFP-construct). This is of course much easier in transformed cell lines than in primary cells, which often don't like being virally transduced. We acknowledge the limitation and, during the revision, we have invested a lot of work into getting the auxin-dependent CAD into BJ and WI-38 cells. It proved technically very challenging because the cells grow very poorly upon transduction. In the end, we did manage to establish WI-38 cells carrying an auxin-inducible CAD-construct. Within the time we had, we could not do all the experiments we have done for the other models but it appears to be clear that CAD-activation is sufficient to induce senescence also in these cells, even though we could perform only two experiments in the time we had (Appendix Figure S3).

For all the work, we are grateful to the reviewer for his/her insistence on primary cells. This makes the study much stronger.

6. In the discussion section, the statement, "It was surprising to observe that doxorubicin, which intercalates in and directly damages DNA also depends on CAD for senescence induction," might be an overgeneralization. It would be beneficial to investigate whether CAD is necessary for DNA damage-induced senescence in primary cells by using gamma or X-ray irradiation and treatments with agents like etoposide or doxorubicin on primary fibroblasts.

We have to agree with this criticism. We used both H₂O₂ and doxorubicin as directly DNA-damaging agents and obtained variable results. As suggested, we did the experiment with doxorubicin on WI-38 cells, and there was only a very small effect of CAD-deficiency (Fig. EV3). As the reviewer suggests, there appears to be variability. We did not have the time to do extensive experiments changing conditions but we believe it possible that it will depend on concentration and kinetics of treatment (in addition to the time point). We had already suggested in the original manuscript that it may be that CAD can contribute because it is activated by DNA-damage and ATM/ATR (as described in the Larsen paper). If that was correct, there may be a threshold above which no CAD will be required. We believe that it is clear now that it is indeed the DNA-damaging activity of CAD that is required for senescence, which can be bypassed by directly DNA-damaging agents.

7. CCF serves as a marker of senescence and is an integral component in the production of SASP. A study by Vizioli et al. (2020) in *Genes and Development* proposed that the Mitochondria-ROS-JNK signaling pathway drives CCFs and the SASP. Another research by Ichim et al. (2015) in *Molecular Cell* demonstrated the requirement of CARD for JNK1/2 activation and subsequent γ H2A.X induction. However, the question remains whether CARD is essential for CCF formation. The authors mentioned, "We and others have reported previously that CAD activation can generate CCF. DNA-recognition downstream of CAD activation seems a likely contributor to the generation of the SASP and the induction of senescence." This assertion is based on extrapolation without specific references to CCF. The given references did not discuss about CCF formation. The confusion might stem from the distinction between Micronuclei (MN) and CCF. These events have different characteristics and mechanisms of formation, despite sharing common molecular markers. MN formation occurs in the context of cell division and is linked to chromosome missegregation, while CCF formation is associated with the loss of nuclear membrane integrity in senescent cells (Miller et al., 2021, *Cell*).

This is a valid comment. Indeed, we have been somewhat negligent with the nomenclature. We (PMID: 35393399) and others (PMID: 25702873) have reported that CAD-activity can generate micronuclei. The role of DNA-damage in the generation of CCF is not as clear but their positivity for γ H2AX suggests that DNA-damage also plays a role in their generation (as discussed in the Miller-paper). We have now stained for the histone markers typically associated with CCF but not micronuclei (H3K9me3, H3K27me2me3) upon CAD-activation and find cytosolic DNA-structures that are negative (probably micronuclei) and such that are positive (likely CCF) (Appendix Figure S3). CAD therefore appears to be able to cause the appearance of both micronuclei and CCF.

Referee #2

The manuscript by Haimovici et al. aims to demonstrate that the cellular endonuclease Caspase-Activated DNase (CAD) is an important factor in senescence induction. The authors claim that CAD activation by sub-lethal signals in the apoptosis pathway is responsible for the initiation of senescence. Additionally, they find that various inducers of cellular senescence, such as oncogenic RAS, interferon and doxorubicin treatment, depend on CAD in their capacity to trigger senescence. Finally, the authors tested the contribution of CAD to senescence in various mouse tissues and found reduced senescence in organs of aged CAD-deficient mice. Unfortunately, the study is too preliminary in nature and most (if not all) of the claims are devoid of robust experimental support. Importantly, there is a complete lack of mechanistic insight into how CAD may activate senescence-related factors. In general, the manuscript lacks depth and the methodology is flawed in many parts.

We thank the reviewer for his/her comments on the manuscript. Although we do not know how CAD-dependent DNA-damage induces senescence, we believe that our findings are an important advance. Until now, we had known that experimental DNA-damage can trigger senescence and that DNA-damage is associated with senescence but not what is the cause of DNA-damage induced outside experimental DNA-damage or whether DNA-damage is really essential and what are the mechanisms involved. With our findings we can only partially address how DNA-damage leads to senescence (see below for the SASP) but we identify the cause of DNA-damage both *in vitro* for multiple stimuli and *in vivo* during ageing. The association of DNA-damage with senescence has long been mysterious, and we identify a very relevant source of this damage.

Some specific remarks:

1. In Figures 1A-B the authors assess induction of senescence in MDA-MB-231 breast cancer cells upon ABT treatment (inactivating anti-apoptotic factors) in Control (Ctrl) and CAD-deficient cells. The authors' approach relies on estimation of SA- β -Gal levels and the expression of a few other markers. However, this approach is not sufficient any more in establishing the presence of senescence in any biological sample or cell culture. As explained in the senescence detection algorithm established by Kohli et al. (2021), SA- β -Gal or lipofuscin levels should be carefully quantified, along with expression of proliferation markers (such as Ki-67) and major senescence-related markers (p16, p21 and lamin B1). The approach followed by the authors is incomplete as they do not present changes in proliferation nor expression of key senescence markers (p16, lamin B1). This is an important omission observed throughout the manuscript (e.g. Fig. 2A-B and H-I, 3A etc), therefore it is impossible to determine whether senescence is correctly estimated in each case.

We fully agree that senescence detection is not straightforward and that several markers have to be measured. If we did not always test for all markers in all settings it was because we do mostly not establish novel situations of senescence but show that senescence-induction by established stimuli depends on CAD. We acknowledge the validity of the comment and as suggested, we have now extended the analyses and add the generally accepted markers (also mentioned by the reviewer) in the situations we test (please note that we had tested initially for p16; we had referred to it as CDKN2a, which is not unusual and commonly done. We now call it P16INK4A throughout). We have added the analysis of lamin B1 and of proliferation (Edu-incorporation; Ki67 *in vivo*) where we had not had it; the analysis now includes p16, p21, lamin B1, proliferation, β -Gal, morphology and SASP-genes for most cases. The results have been added to most figures in the manuscript.

2. In Figures 1A-B, there is no data showing CAD expression in Ctrl versus CRISPR/Cas9-engineered CAD-deficient cells. This is an important control that is obviously missing. Again, in Figures 2E-G the authors do not provide any validation of CAD levels in CAD-deficient MEFs compared to controls. All these data have to be presented (for CRISPR/Cas9 experiments sequencing results have to be shown and for the mouse-derived MEFs genotyping data should be presented).

We have done these controls and now add the data for CAD-deficiency in MDA-MB231 as well as the newly added BJ and WI-38 cells (please see new Fig. EV1). The CAD-deficient mice have been characterized and published in 1998 (as referenced in the manuscript). We now also include the results from the genotyping of the MEFs, following the original protocol (Fig. EV1).

3. Many parts of the manuscript are highly speculative. For example, in page 5 the authors mention "We further exposed MEFs to H₂O₂, which directly induces DNA-damage and therefore bypasses the activation of CAD by caspases". Where do the authors base this assumption on? References and/or experiments are definitely required to support this statement.

We have now added references. Direct DNA-damage by H₂O₂ is so well established that we didn't include a reference initially. One instance is this 1988 paper showing DNA-damage by H₂O₂ in *E. coli* in Science (PMID: 2834821), although the understanding has evolved through

numerous publications since (one example is PMID: 29232557). We have added these citations to the manuscript.

4. The authors claim that the data in Fig. EV1 indicate the principal ability of CAD-deficient cells to undergo senescence upon DNA damage. Unfortunately, the displayed q-PCRs in Fig. EV1 are insufficient to support the claim. A series of well-designed experiments with appropriate controls with regards to senescence (see point 1 above) would be, instead, required to demonstrate the role of CAD in senescence upon DNA damage and/or oxidative stress.

We have extended these analyses and now show staining for β -Gal, proliferation (Edu), lamin B1 protein, two senescence-associated genes (p16, p21) and one SASP factor (IL-6) (Appendix Figure S2). The results indicate that H₂O₂ can, presumably through direct DNA-damage, induce CAD-independent senescence.

5. The reliance of senescence induction on CAD upon doxorubicin treatment is not adequately investigated and the authors only provide speculations to try to interpret the results. A better explanation should be experimentally provided regarding the potential differential requirement of CAD upon H₂O₂ or doxorubicin-induced DNA damage.

This is certainly a relevant point. Like H₂O₂, doxorubicin can directly damage DNA and should therefore cause senescence CAD-independently. On the other hand, a recent report shows that DNA-damage itself, through ATM/ATR, can activate CAD (PMID: 35482866). This may indicate – as we suggest in the manuscript – that CAD may contribute but there may also be conditions where senescence is induced by these agents through CAD-independent DNA-damage. We have now included another cell type, WI-38 fibroblasts, and here the contribution of CAD is much less (Fig. EV3). Please see the comments on CAD and H₂O₂ above (Appendix Figure S2). We propose the explanation, which we believe is very likely correct: DNA-damage can induce senescence (this is well established); DNA-damage in many situations is the result of CAD activity; following direct DNA-damage there may be a variable contribution of CAD, which in this situation is likely activated by ATM. The data now show more clearly that it is indeed the DNA-damaging role of CAD that is required for senescence induction, and that this role can be bypassed by direct experimental induction of DNA-damage.

6. In Figure 3B, why do the authors assess only IL-6 levels in the supernatant of MEFs in regards to SASP? Deriving a complete SASP profile would be much more informative and absolutely required to make statements regarding the SASP.

We agree. We had done the IL-6-measurements by ELISA. We have now tested SASP-components by bead array and include IL-6, CCL2, CCL5, CXCL1 and CXCL10 (Fig. 3); TNF, MMP3 and MMP13, other SASP-factors are shown at the mRNA-level.

7. In Figures 4E-G the authors show that incubation of BJ fibroblasts with medium from HaCaT cells expressing active CAD leads to BJ reduced proliferative capacity (estimated by EdU incorporation), signs of a DDR and loss of lamin B1. However, showing only EdU incorporation changes (which appear also statistically insignificant) and percentages of H2AX+ cells without any representative image of relevant stainings does not convincingly indicate DDR. Surprisingly, loss of lamin B1 is never shown, but the authors rather refer to it as a fact. In general, the results are too weak to support most of the statements in the manuscript.

We have redone the EdU experiments; now (including the new results) the difference between control treatment (DMSO) and auxin-treatment is statistically significant (Fig. 4F). We also include representative images of the γ H2AX-stain as requested (Fig. EV4). We further include the lamin B1-Western blot (Fig. 4). We had however already included the lamin B-stain by microscopy, showing the total fluorescence on a per cell basis (as is commonly done, see for instance PMID: 28759028); on this basis we had made the statement. There is a highly significant loss of lamin B1 staining (Fig. 4H).

8. *The claim that CAD induces DDR at telomeres is already known and described, also according to the authors (ref. 29 in the manuscript). Additionally, Figure 4H has two serious problems: no control is shown and only one nucleus is presented. No conclusions can be safely drawn, but there seems to be little point in doing those experiments as the findings are no longer novel. The potential novelty of the study is somewhat compromised by previous findings.*

As the reviewer states, we have acknowledged the previous report in our manuscript. We can actually see no fault in confirming the earlier finding that CAD induces a DDR at telomeres when we study the induction of senescence by CAD. We do find it difficult to follow the argument that this minor point, where we confirm previous results, compromises the novelty of the study. The results are shown in Fig. EV4. We now also show a number of nuclei and include the control sample as suggested (Fig. EV4B).

9. *For in vivo data, senescence validation experiments need to be appropriately repeated (see point 1 above). In Figure 5E, extremely high levels of senescence are displayed in wt tissue, which does not appear reasonable as the intestine is known for displaying high renewal rates, and may be attributed to non-specific SA- β -Gal absorption by the mucous. In any case, the displayed results are not sufficient to strongly support the claims put forward by the authors. A much more thorough validation with updated senescence detection tools and assays is required.*

We fully agree that senescence detection has to follow the proper algorithm. For *in vivo*-studies in mice, a landmark paper is the analysis done in a publication in *Nature* (Baker et al. 2016, PMID: 26840489); this study identified senescence by SA- β -Gal staining and mRNA expression of several senescence markers. We followed the same approach for our study.

For the kidney, we also show loss of lamin B1 in wt but less so in CAD-deficient mice; we have now added Ki67-staining in the kidney, which confirms the result (more positive cells in the old CAD-deficient mice).

We are not sure why the reviewer is of the view that our data on the intestine – which are controlled by CAD-k.o. – are unreliable. Indeed, the high division rate of intestinal mucosa may lead to high levels of senescence. The same effect is seen in the organoids from wt and CAD-deficient mice (now Fig. EV5A-C). We now further add Ki67-staining (as well as laminB1) of the intestine, which also gave the very clear result of more proliferation in the CAD-deficient cells (new Fig. 5E).

In all four different organs we have tested we have thus found evidence of reduced senescence in CAD-deficient cells by a number of parameters.

10. *In their Discussion (page 10), the authors mention an interesting point, on which their whole manuscript is based: that activation of CAD within a certain signaling threshold may lead to senescence, otherwise other processes such as apoptosis may be induced. However, this notion is not experimentally tested at all and remains purely theoretical. Mechanistic insights to justify a potential interplay between CAD and known molecular determinants of senescence are completely absent.*

We have taken this up and tested it further with three lines of primary fibroblasts (BJ, WI-38 and MEFs). We used ABT-737, a BCL-2/BCL-X_L-inhibitor: we know exactly how this molecule works and how it triggers apoptosis signals. There are very many studies showing that ABT-737 induces apoptosis in a concentration-dependent manner. ABT-737 activated CAD in a caspase-dependent fashion in the conditions we used (Fig. EV1C) while little or no apoptosis was induced (new Fig. EV1B); the signals are sub-lethal. These conditions induced senescence in all three cells (Fig. 1, Fig. EV2; and in MDA-MB-231, as already in the original manuscript, Fig. EV2). The results provide the experimental test that sub-lethal activation of CAD promotes senescence. The mechanistic link to the mitochondrial apoptosis system and caspases upstream of CAD is therefore clear.

We have also looked into a potential downstream effect of CAD and focussed for this on a link to cGAS. cGAS has initially been found to be essential for senescence in a number of

settings (PMID: 28533362; PMID: 28759028). cGAS has recently further been shown to be required only for the SASP but not for other features of senescence upon irradiation (PMID: 37821702). Because CAD can generate cytosolic fragments that can recruit and activates cGAS (PMID: 35393399), we tested whether CAD-effects were transmitted through cGAS. We deleted cGAS in the HaCaT cells where auxin activates CAD. We made the observation that deletion of cGAS did not alter the senescence-associated features of β -Gal-expression, lamin B1-loss, reduction in proliferation (and led to a slightly reduced expression of p21) but had a very strong effect on the induction of IFN- β and CXCL10, and a (less pronounced but significant) reduction in the secretion of IL-6 (Fig. 4). This suggests that some of the effects of CAD-activation require cGAS – in particular the SASP – while others do not.

Referee #3

Based on description of sub-lethal caspase activity leading to CAD (DNase) activation made by this lab and others, Haimovici and colleagues investigated a role for CAD in senescence. Summarizing the paper, the authors demonstrate a convincing role for CAD following different senescence triggers (DNA-damage, OIS, replicative) and demonstrate that CAD activation is itself sufficient to cause senescence. This exciting work makes an important contribution to emerging data linking apoptotic signaling and senescence. The data largely support the authors' conclusions. That said, I have some points that should be addressed.

Thank you for the thorough scrutiny and the comments. We have addressed the points raised.

- trying to get a sense of the absolute requirement of CAD in senescence (vs contributory factor), in the context of OIS, dox and replicative senescence shown here, do the CAD deficient cells eventually undergo senescence to the same degree as WT cells or is there always a persistent reduction (in as far as the authors have analyzed)?

This is an interesting observation. We had not tested this for later time points than we show in the paper. For the replicative senescence we have waited until the wt had overcome its crisis and resumed proliferation. It seems reasonable to assume that there would be nothing later on (we have not documented this carefully but we have used the 3T3 MEFs in numerous experiments and have not observed a growth arrest in CAD-deficient cells that might indicate that the cell later undergo senescence). We have now tested this question specifically for OIS, using ras as before. We had to switch to BJ fibroblasts because the MEFs we had used initially enter replicative senescence too quickly, which would likely interfere. While there was basically no indication of senescence in CAD-deficient MEFs at 10 days (already in the first version of in the manuscript, Fig. 2), at 14 days a reduction in proliferation was seen in BJ cells, and at 21 days it had reached the level of wt cells (Appendix Figure S1). At least some effects therefore appear at least in some cells to occur also in the absence of CAD. Thank you for suggesting this.

- as the authors note the role of CAD in doxorubicin mediated senescence is somewhat surprising given that DOX is a potent DNA-damage inducer, as discussed, this raises the possibility that CAD activation independent of caspase activity following DNA-damage (as described in a recent Science paper) is mediating senescence in this context. This should be directly tested, one way would be to overexpress non-caspase cleavable ICAD in these cells, asking do they undergo senescence to the same degree - if so, demonstrates non-caspase dependent activation of CAD in the context of doxorubicin mediated senescence, which is an important message.

We agree that this is an interesting point, also mentioned by reviewer #2 (above). We are a little hesitant with the uncleavable ICAD. We have expressed CAD and ICAD in a number of situations and have seen surprising results in expression levels, presumably because the two appear to act as co-chaperones during translation. We therefore used caspase-inhibitor (this is also what the authors did in the Science paper, PMID: 35482866). An alternative explanation to CAD-activity is that ATM/ATR (see that paper) activate CAD. We tested expression of p21 and found no effect of caspase-inhibitor; we also tested ATM-inhibitor,

which also had no effect (Fig. EV3). We are a little unsatisfied with the result because the cells started looking a little unhappy after the week of inhibitor treatment but we did not have the time to do extensive kinetic experiments. We discuss this cautiously in the manuscript but it does appear to show that caspases are not required. We have also repeated the doxorubicin-experiment with WI-38 fibroblasts (also a suggestion by reviewer 1). In those cells, CAD makes a much smaller contribution than in MDA-MB231 (which we had used initially). It does seem like DNA-damage by doxorubicin can work CAD-independently, although a secondary activation of CAD by DNA-damage may play a role in some conditions.

- given the evident negative role of senescence burden impacting longevity and/or health, can the authors comment on the phenotype of aged CAD deficient mice, for instance, do they display any evident extension of lifespan and/or health?

We have started looking into this. We have so far identified a life-span effect in heterozygous but not homozygous CAD-deficient animals, as well as reduced anxiety and less fat accumulation in mutant mice. All of these are signs of reduced organismal ageing. We would prefer not to include the data in the figure because proper assessment will require substantial additional experimentation. We mention the results in the manuscript though as suggested by the reviewer.

- in terms of analysis of senescence, it would be helpful if images of cells in Figure 1 and 2 (ABT-737, OIS and dox) are included and commented on, do the cells display typical senescent morphology? Secondly, do the B gal positive cells display cell cycle arrest?

We now show pictures of the typical morphology (Fig. 1, EV2, Appendix Figure S2). We have separately tested for growth arrest/proliferation (Fig. 1, 2, 4, 5, EV2, EV3, EV4, Appendix Figure S1 and S2 using Edu; Ki67 *in vivo*). It is difficult to test this in the same cells but the analyses are we believe quite clear.

- for the CRISPR lines, blots demonstrating CAD deletion should be included.

We have included the blots, including the new data with the primary cells (Fig. EV1A).

- it would be helpful if the authors could discuss their findings in the context of other recent publications linking apoptotic signaling and senescence, please see PMID 36414711, 37821702.

We agree (one of the studies was published after submission of our article). We have included a discussion of these studies.

Dear Georg,

Thank you again for submitting your revised manuscript to The EMBO Journal and for your patience during peer review. We have received the comments of the three referees that were asked to re-assess your study (included below), and I have already shared them with you. I would also like to thank you for your responses to the remaining referees' points, which were very helpful for us to reach a balanced decision without further delays.

Referees #1 and #3 are satisfied with the revision mentioning that their previous concerns have been comprehensively addressed, and they now recommend publication of the manuscript as is. Referee #2 also recognizes the improvement of the manuscript, but still identifies three points that have not been completely addressed.

In light of the referees' input and your responses to the remaining concerns, I would like to inform you that we can move forward with your manuscript for publication in The EMBO Journal without requesting additional experimental work. However, we do request you to provide us with a detailed point-by-point response to the remaining concerns -along the lines of the draft responses you have already sent me- to be included in the peer review file that will be published along with your article. Please cite properly your other publication in print where you show the data regarding the point #3 of referee #2 (this could also be done later -but before online publication of your article- if you do not have a citable DOI yet).

There are also a few minor changes and corrections that we need from you to make in a revised version of your manuscript before we can proceed with its acceptance:

- You can now remove previous highlighting and "tracked" changes from your manuscript file.
- Please provide a list of up to 5 keywords after your Abstract in the revised manuscript.
- Please change "Material and Methods" to "Materials and Methods".
- We noticed that no grant number is provided for your funding source in the Acknowledgements section. If such a number is available, please add it to both the revised Acknowledgements section and to the online manuscript handling system during submission of your revised manuscript.
- The author contributions statement should be removed from the manuscript file. Instead, we now use CRediT to specify the contributions of each author in the journal submission system. Please use the free text box to provide more detailed descriptions during submission. See also our guide to authors for more information:
<https://www.embopress.org/page/journal/14602075/authorguide#authorshippinguidelines>.
- Please rename the heading of your competing interests statement to "Disclosure and competing interests statement".
- Please note that the literature citations in your References list should not be numbered; instead, they should be provided in alphabetical order, and "et al." should follow the names of the first 10 authors in case of publications with more than 10 co-authors. For more information on our reference format please visit:
<https://www.embopress.org/page/journal/14602075/authorguide#referencesformat>.
- Please add page numbers to the Table of Contents on the first page of your Appendix.
- As per our journal's policy, "data not shown" (stated on page 15 of your manuscript) is not permitted. All data referred to in the paper should be displayed in the main or Expanded View figures, or in the Appendix. Please add these data or change the text accordingly if these data are not central to the study and its conclusions, or properly cite the respective published sources if these data can be found elsewhere.
- We noticed that Figure callouts are missing for panels A-H of Figure 1; Fig 2H; Fig 3B; and Fig 5F. Please make sure that all Figure panels are called out (in alphabetical order) in your revised manuscript.
- Fig EV2 is called out before Fig EV1A,C. Please make sure that all Figures (and their panels) are called out in alphabetical order in your revised manuscript.
- Please add the heading "Figure legends" before the main Figure legends.
- Please rename the Expanded View Figures to "Figure EV1" etc. in their legends.
- Please indicate the statistical test used for data analysis in the legends of Figures EV 5a-c, e-f.

- Please note that information related to n is missing in the legends of Figures EV 4d; EV 5c.
- Please note that the error bars are not defined in the legends of Figures EV 1b; EV 4c-d; EV 5a-c, e-f.
- Please note that the scale bar is missing in the legends of Figures 5b, e.
- Please note that the scale bar and their definition are missing for Figures 1a, e; 3a; 4b; 5a-b, d-e; EV 2a, e; EV 3a, d; Ev 4a-b; EV 5a.
- Please note that in Figures 5b, e; EV 4a-b; the scale bar unit should be corrected from μM to μm (both in the Figure legend).
- Please note that the asterisks are not defined in the legend of Figure EV 4c. This needs to be rectified.

Please also note that as part of the EMBO publications' Transparent Editorial Process, The EMBO Journal publishes online a Peer Review File along with each accepted manuscript. This File will be published in conjunction with your paper and will include the referee reports, your point-by-point response and all pertinent correspondence relating to the manuscript. You can opt out of this by letting the editorial office know (contact@embojournal.org). If you do opt out, the Peer Review File link will point to the following statement: "No Peer Review File is available with this article, as the authors have chosen not to make the review process public in this case."

We look forward to seeing a final version of your manuscript as soon as possible. Please use this link to submit your revision: <https://emboj.msubmit.net/cgi-bin/main.plex>

Best regards,

Ioannis

Referee #1:

The authors have performed an enormous amount of work to address my concerns. They have addressed almost all of my concerns, and I am satisfied. I recommend this manuscript for acceptance.

Referee #2:

In their revised manuscript, Haimovici et al. have addressed some of the concerns previously raised. In particular, as requested, the authors include additional markers to confirm the establishment of senescence whenever this is the case, which strengthens the conclusions. However, there are a few remaining points:

1. The observations in WI-38 cells do not resonate with the overall hypothesis as most of the tested markers yielded non-significant differences between Ctrl and CAD-deficient cells. On the other hand, the observations in MEFs (Fig. 1 and 2) seem to be more aligned with the model put forward by the authors. Lamin B1 blots appear not to have worked for MEFs (Fig. 2C and G).
2. In revised Fig. 3C, the authors indeed included more SASP-related markers, but they (e.g. CCL2, CCL5) do not follow the anticipated pattern regarding induction of senescence in wt and CAD -/- MEFs. In general, the SASP profiling presented by the authors remains incomplete.
3. cGAS experiments provide some mechanistic insight into CAD-mediated senescence, however no control of cGAS depletion is provided in the knockdown experiments.

Referee #3:

The authors have comprehensively addressed all my comments.

Point-by-point-reply

Referee#2 (Report for authors)

In their revised manuscript, Haimovici et al. have addressed some of the concerns previously raised. In particular, as requested, the authors include additional markers to confirm the establishment of senescence whenever this is the case, which strengthens the conclusions. However, there are a few remaining points:

We thank the reviewer for his/her efforts in assessing our manuscript, and we are very pleased that he/she acknowledges the amendments and additions we have made. We will respond to the points below.

1. The observations in WI-38 cells do not resonate with the overall hypothesis as most of the tested markers yielded non-significant differences between Ctrl and CAD-deficient cells. On the other hand, the observations in MEFs (Fig. 1 and 2) seem to be more aligned with the model put forward by the authors. Lamin B1 blots appear not to have worked for MEFs (Fig. 2C and G).

The first point is about significance. The reviewer is correct in that we have in some WI-38 panels reported p-values of for instance 0.06 (Fig. 1A, Edu+ cells; Fig. 1C, p=0.0588, P16). In some cases, we report non-significant differences between a marker of senescence in wt and CAD-deficient cells (Fig. 1A, b-Gal+ cells). This point is not really necessary; we could also compare CAD-k.o. untreated and ABT-737-treated and say the difference is non-significant (which is all up a little less honest because lack of significance could also be founded in greater variation). Other markers, also for WI-38 cells, in fact do have a p-value <0.05 (Fig. 1D). In our view, the point we make is quite clear, and it is now supported by results in other cell lines (MEFs, Fig. 1E-H; BJ fibroblasts and MDA-MB231 in Fig. S2). As we all know, the p-value of 0.05 is a convention, and this way of assessing significance of experimental data has weaknesses. We would like to suggest that it is more important, and in our view convincing, that we have now consistent evidence from four different cell types.

We are not sure that one can say that the LaminB1-blot did not work on the basis that we did not observe a loss of it by the senescence-inducing stimulus. It is quite clear in the senescence field that senescence markers are very variable, may change over time and do not all have to be present in one given situation (please see for instance PMID: 28844647 and 31675495). This is a quote from the first of these two papers: '*The senescence program is **variably characterized by several non-exclusive markers**, including constitutive DNA damage response (DDR) signaling, senescence-associated b-galactosidase (SA-bgal) activity, increased expression of the cyclin-dependent kinase (CDK) inhibitors p16INK4A (CDKN2A) and p21CIP1 (CDKN1A), increased secretion of many bio-active factors (the senescence-associated secretory phenotype, or SASP), and reduced expression of the nuclear lamina protein LaminB1 (LMNB1)*'. The reviewer made the point of looking at several markers very clearly in his/her first assessment, and we have done very extensive additional experimentation to look at numerous markers (which has undoubtedly improved the manuscript). However, it is simply the biology that senescence does not always have all markers. Further, as we had said earlier, the situations we test are not novel situations of senescence that would require extensive proof of senescence – what is novel is that CAD is required. Therefore, this point seems not essential, in particular in the light of all the additional evidence we provide.

2. In revised Fig. 3C, the authors indeed included more SASP-related markers, but they (e.g. CCL2, CCL5) do not follow the anticipated pattern regarding induction of senescence in wt and CAD -/- MEFs. In general, the SASP profiling presented by the authors remains incomplete.

We tested five SASP-factors in the supernatants and found three significantly induced by the protocol of replicative senescence. Of these, two significantly and very clearly depended on CAD, one did not. Two factors were not actually induced by the senescence-protocol. Fig. 3B shows another SASP factor under the same conditions but on the mRNA-level (TNF), which also was induced in wt but not CAD-deficient cells. We find similar results for four more senescence-associated (non-SASP) genes (mRNA), a phenotype in proliferation and in beta-Gal-expression in these cells. We truly cannot see how this is not convincing. Again, SASP-secretion is variable and may even change over time (see for instance PMID:37445973). We investigate one time point where we find the results we show. Even though the reviewer is correct in stating that more factors could be investigated, the point we make (less senescence in CAD-deficient cells) to us seems very clear; the other two reviewers appear to agree.

3. cGAS experiments provide some mechanistic insight into CAD-mediated senescence, however no control of cGAS depletion is provided in the knockdown experiments.

We apologize for this omission. We have also used the cells in another paper that is in print at Cell Death and Differentiation (in that paper, we investigate the activation of CAD during infection) and show the blot there. The reference (Moeed, Thilmany et al. 2024) has been properly cited in the text and in the references section in the revised manuscript. You can find the citation in the results section ("CAD activation is sufficient to induce senescence", page 9).

As said above, this as well as the other reviewers have very substantially contributed to the improvement of our manuscript, and we are very grateful for their help. We do hope that these explanations make the manuscript acceptable for publication.

Dear Georg,

Thank you for resubmitting your revised manuscript and addressing most of our editorial requests. However, there are still a few issues that remain to be addressed before we can proceed with acceptance of your article for publication in The EMBO Journal.

In particular:

1. We noticed that the Data availability statement -which is mandatory and was present in the previous version of your manuscript ("Data availability: This study includes no data deposited in external repositories.")- has now been removed from the uploaded version. Please note that this statement must be included and make sure that different versions of the manuscript have not been inadvertently mixed up.

2. The following requests for corrections or clarification in your Figure legends have not been addressed (or have been addressed insufficiently). Please make sure that each one of the following requests is completely and clearly addressed. I would like to note that we cannot proceed with publication of the manuscript before these points are adequately answered:

- Please indicate the statistical test used for data analysis in the legends of Figures EV 5a-c, e-f. If use of a specified test applies to more than one panels, please state so explicitly in the legend.

- Please note that information related to "n" is missing in the legends of Figures EV 4d; EV 5c. Please provide the numbers for each panel, even if the results shown in a panel relate to experiments shown in the other panels of the same Figure.

- Please note that the error bars are not defined in the legends of Figures EV 1b; EV 4c-d; EV 5a-c, e-f.

- Please note that the scale bar and their definition are missing for Figures 1a, e; 3a; 4b; 5a-b, d-e; EV 2a, e; EV 3a, d; Ev 4a-b; EV 5a.

- Please note that the asterisks are not defined in the legend of Figure EV 4c. This needs to be rectified.

We look forward to seeing a final version of your manuscript as soon as possible. Please use this link to submit your revision:
<https://emboj.msubmit.net/cgi-bin/main.plex>

Best regards,

Ioannis

All editorial and formatting issues were resolved by the authors.

Dear Georg,

Congratulations on an excellent manuscript, I am very pleased to inform you that it has been accepted for publication in The EMBO Journal. Thank you for your comprehensive responses to the referee concerns. There are only a few minor textual edits I would like to suggest before publication; I will shortly send you another message with them for your review.

Your manuscript will now be processed for publication by EMBO Press. It will be copy edited and you will receive page proofs prior to publication. Please note that you will be contacted by Springer Nature Author Services to complete licensing and payment information.

If you have any questions, please do not hesitate to contact the Editorial Office. Thank you for your contribution to The EMBO Journal. It has been a pleasure working with you!

Best wishes,

Ioannis
